# CPPO: Accelerating the Training of Group Relative Policy Optimization-Based Reasoning Models

**Zhihang Lin**[1,2], **Mingbao Lin**[3], **Yuan Xie**[2,4]*, **Rongrong Ji**[1†]

[1]Key Laboratory of Multimedia Trusted Perception and Efficient Computing,
Ministry of Education of China, Xiamen University, 361005, P.R. China
[2]Shanghai Innovation Institute, China
[3]Rakuten, Singapore
[4]East China Normal University, Shanghai, China
lzhedu@foxmail.com, linmb001@outlook.com, yxie@cs.ecnu.edu.cn, rrji@xmu.edu.cn

## Abstract

This paper introduces Completion Pruning Policy Optimization (CPPO) to accelerate the training of reasoning models based on Group Relative Policy Optimization (GRPO). GRPO, while effective, incurs high training costs due to the need to sample multiple completions for each question. Our experiment and theoretical analysis reveal that the number of completions impacts model accuracy yet increases training time multiplicatively, and not all completions contribute equally to policy training—their contribution depends on their relative advantage. To address these issues, we propose CPPO, which prunes completions with low absolute advantages, significantly reducing the number needed for gradient calculation and updates. Additionally, we introduce a dynamic completion allocation strategy to maximize GPU utilization by incorporating additional questions, further enhancing training efficiency. Experiments show that CPPO achieves up to $7.98\times$ speedup on GSM8K and $3.48\times$ on Math while preserving or even enhancing the accuracy compared to the original GRPO. We release our code at https://github.com/lzhxmu/CPPO.

## 1 Introduction

Recently, there has been a surge in the development and application of advanced reasoning models, with models such as OpenAI-o1 [11], Deepseek-R1 [7], and Kimi-k1.5 [25] being prime examples. These models exhibit remarkable capability in complex reasoning tasks, such as mathematics, coding, and scientific reasoning through step-by-step inference and reflection.

Reinforcement learning has been proven to be an effective method for training reasoning models. Deepseek-R1 [7] demonstrates that reasoning patterns can be effectively elicited through rule-based reinforcement learning. It employs Group Relative Policy Optimization (GRPO) [21], which differs from Proximal Policy Optimization (PPO) [20] by estimating the baseline directly from group scores, eliminating the need for a critic model. However, this necessitates sampling a group of completions for each question, rendering the training process computationally expensive. Subsequently, GRPO computes the reward for each completion using a rule-based reward function and calculates the relative advantage of each completion. To ensure training stability, GRPO also calculates the ratio of the predicted probabilities of the policy model, reference model, and old policy model for a group of completions as part of the policy objective function, further increasing the training overhead of reinforcement learning. The substantial training overhead of GRPO limits its training efficiency and scalability. Improving the training efficiency is an important and practical problem.

---

*Project Leader
†Corresponding Author

39th Conference on Neural Information Processing Systems (NeurIPS 2025).

The computational expense of GRPO training primarily stems from its core design: generating a large group of completions per prompt for intra-group comparison, which makes the training process computationally expensive. Moreover, the forward computation of GRPO scales by a factor of $(3\times)$ completion number. It is natural to question whether the contribution of each completion to the reinforcement learning process is equal. In Sec. 3.2, we find that the contribution of each completion is related to its relative advantage. In other words, the contribution of each completion to the policy model training is not equal. This insight inspires us to accelerate GRPO by pruning completions.

In this paper, we propose Completion Pruning Policy Optimization (CPPO) to accelerate Group Relative Policy Optimization (GRPO). Given that each completion's contribution to the reinforcement learning process varies significantly and is closely related to its relative advantage, our CPPO prunes completions based on advantage, thereby accelerating the reinforcement learning process. Specifically, the policy model initially samples a group of completions for each question. Subsequently, the relative advantage of each completion is computed via the reward function. CPPO then prunes completions with low absolute advantage values, retaining only those with high absolute advantage for loss computation. This process considerably reduces the number of completions needed for training, thus speeding up the training process. Moreover, we observe underutilized GPU resources due to completion pruning, leading to resource waste. To tackle this, we introduce a dynamic completion allocation strategy that fills each device with completions from new questions, fully utilizing GPU resources and further enhancing training efficiency.

We have conducted experiments on multiple challenging benchmarks and models of different scales to evaluate CPPO's effectiveness. Specifically, we train the Qwen-2.5 series models [29], such as Qwen-2.5-1.5B-Instruct and Qwen-2.5-7B-Instruct, on math datasets including Math [8] and GSM8K [4]. The results demonstrate that CPPO achieves up to $7.98\times$ speedup on GSM8K and $3.48\times$ on Math while preserving or even enhancing the accuracy compared to the original GRPO.

## 2 Related Work

**Large Scale Reasoning Models**. Large Language Models (LLM) [1, 26, 24, 2] have made impressive progress in various natural language processing tasks. Recently, researchers have continued to boost the performance of large language models in reasoning tasks, such as mathematics [4, 8], coding [12], and scientific reasoning [19]. Snell *et al.* [23] use dense, process-based verifier reward models and adaptively update the model's response distribution based on the test-time prompt to enhance reasoning ability. rStar-Math [6] proposes a self-evolved deep thinking approach that significantly boosts the math reasoning capabilities of small LLMs. OpenAI-o1 [11] uses large scale reinforcement learning to train a reasoning model that can solve complex reasoning tasks, achieving state-of-the-art performance on multiple benchmarks. However, the training details of OpenAI-o1 have not been released, making it difficult to replicate and expand the reasoning model.

**Reinforcement Learning**. Recently, DeepSeek-R1 [7] has incentivized the reasoning capability of large language models through Group Relative Policy Optimization. Inspired by DeepSeek-R1's success, Logic-RL [28] adopts the REIFORCE++ algorithm to enhance the training efficiency and stability of rule-based reinforcement learning. Hu *et al.* [10] demonstrate that the vanilla PPO algorithm, without KL divergence constraint, is sufficient to scale up both response length and benchmark performance on reasoning tasks. Nevertheless, these reinforcement learning algorithms universally require multiple completions for each question, resulting in substantial computational costs. There is an urgent need to accelerate the training of reinforcement learning algorithms.

**Inference Acceleration for Reasoning Models**. The enhancement of model inference capabilities is often accompanied by increased computational overhead and longer response times. Recent works have attempted to accelerate the inference process of reasoning models through efficient Chain of Thought (CoT) methods. TokenSkip [27] proposes a controllable CoT compression method that improves reasoning efficiency by selectively skipping less important tokens while preserving critical ones, thus achieving a balance between efficiency and accuracy. Kang *et al.* [13] utilize a compressor to condense an original longer CoT into a shorter one while maintaining key information and interpretability. Although numerous works focus on inference acceleration, the acceleration of reasoning model training remains an underexplored area.

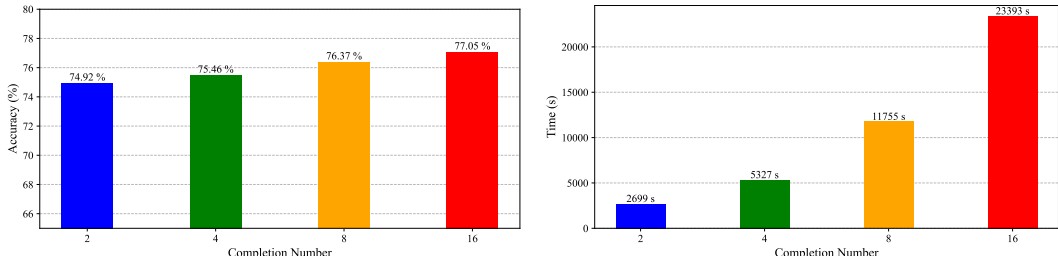

Figure 1: Completion number *vs.* (left) accuracy and (right) training time. Experiments are conducted on GSM8K [4] using Qwen2.5-1.5B-Instruct [29].

## 3 Method

### 3.1 Preliminary

**Group Relative Policy Optimization**. GRPO [7] foregoes the critic model that is typically the same size as the policy model and estimates the baseline from group scores instead. Specifically, for each question $q$ sampled from the dataset distribution $P(Q)$, GRPO generates $G$ completions $\{o_1, o_2, \cdots, o_G\}$ using the old policy model $\pi_{\theta_{old}}$. And then GRPO optimizes the policy model $\pi_\theta$ by maximizing the following objective:

$$\mathcal{J}_{GRPO}(\theta) = \mathbb{E}_{q \sim P(Q), \{o_i\}_{i=1}^G \sim \pi_{\theta_{old}}(o|q)} \left\{ \frac{1}{G} \sum_{i=1}^G \frac{1}{|o_i|} \sum_{t=1}^{|o_i|} \left\{ \min \left[ \frac{\pi_\theta(o_{i,t}|q, o_{i,<t})}{\pi_{\theta_{old}}(o_{i,t}|q, o_{i,<t})} A_i, \right. \right. \right.$$

$$\left. \left. \left. \text{clip}\left( \frac{\pi_\theta(o_{i,t}|q, o_{i,<t})}{\pi_{\theta_{old}}(o_{i,t}|q, o_{i,<t})}, 1 - \epsilon, 1 + \epsilon \right) A_i \right] - \beta \mathbb{D}_{KL}\left[ \pi_\theta || \pi_{ref} \right] \right\} \right\}. \quad (1)$$

where

$$\mathbb{D}_{KL}\left[ \pi_\theta || \pi_{ref} \right] = \frac{\pi_{ref}(o_{i,t}|q, o_{i,<t})}{\pi_\theta(o_{i,t}|q, o_{i,<t})} - \log \frac{\pi_{ref}(o_{i,t}|q, o_{i,<t})}{\pi_\theta(o_{i,t}|q, o_{i,<t})} - 1. \quad (2)$$

Here, $\epsilon$ and $\beta$ are hyperparameters. $\pi_{ref}$ is the reference model which is usually the initial model before reinforcing learning. And $A_i$ is the advantage computed using a group of rewards $\{r_1, r_2, \ldots, r_G\}$ corresponding to the completions within each group:

$$A_i = \frac{r_i - \text{mean}(\{r_1, r_2, \ldots, r_G\})}{\text{std}(\{r_1, r_2, \ldots, r_G\})}. \quad (3)$$

**Rule-based Reward Function**. Instead of training an additional reward model for reward computation, GRPO employs a rule-based reward system that consists of two components:

$$r_i = R_{format}(o_i) + R_{accuracy}(o_i). \quad (4)$$

Here, the format reward $R_{\text{format}}(o_i)$ ensures that the output adheres to the expected structure, while the accuracy reward $R_{\text{accuracy}}(o_i)$ prioritizes correctness with higher reward for accurate responses. The specific reward functions are presented in Appendix A.

**Analyzing Completion Impact on Policy Training**. From Eq. (1), GRPO's training overhead scales linearly with the number of completions sampled per question. This arises from the necessity of calculating predicted probabilities for the policy, reference, and old policy models over all completions. For instance, in DeepSeek-Math [21], using 64 completions requires 192 forward passes per question (64×3), incurring significant computational costs. This raises two critical questions: (1) How does the number of completions affect policy model accuracy? Does increasing completions always enhance performance? (2) Do all completions in a group contribute equally to training?

To address the first question, we conduct an ablation study on GSM8K [4] using Qwen2.5-1.5B-Instruct [29]. Results in Figure 1 show that model accuracy improves with more completions, but training time grows multiplicatively. This indicates diminishing returns on performance gains as

training costs increase. Crucially, reducing completions to cut costs risks degrading reasoning capabilities, making it impractical.

For the second question, we investigate whether completions contribute uniformly to training effectiveness. Our comprehensive analysis in Sec. 3.2 reveals that completion contributions are highly variable, with some samples providing significantly more training signals than others. These findings motivate the development of strategies to identify and prioritize high-value completions, potentially improving training efficiency without compromising model performance.

## 3.2 Completion Contribution Analysis

To measure the contribution of each completion to the policy model training, we first compute the derivative of the policy objective function in Eq. (1) with respect to the model parameters $\theta$ as:

$$
\begin{aligned}
\nabla_\theta J_{GRPO}(\theta) &= \mathbb{E}_{\left[q \sim P(Q), \{o_i\}_{i=1}^G \sim \pi_{\theta_{old}}(O|q)\right]} \left\{ \frac{1}{G} \sum_{i=1}^{G} \frac{1}{|o_i|} \sum_{t=1}^{|o_i|} \left[ \nabla_\theta \left( \frac{\pi_\theta\left(o_{i,t}|q, o_{i,<t}\right)}{\pi_{\theta_{old}}\left(o_{i,t}|q, o_{i,<t}\right)} A_i \right) \right. \right. \\
&\quad \left. \left. - \beta \left( \nabla_\theta \frac{\pi_{ref}\left(o_{i,t}|q, o_{i,<t}\right)}{\pi_\theta\left(o_{i,t}|q, o_{i,<t}\right)} - \nabla_\theta \log \frac{\pi_{ref}\left(o_{i,t}|q, o_{i,<t}\right)}{\pi_\theta\left(o_{i,t}|q, o_{i,<t}\right)} \right) \right] \right\} \\
&= \mathbb{E}_{\left[q \sim P(Q), \{o_i\}_{i=1}^G \sim \pi_{\theta_{old}}(O|q)\right]} \left\{ \frac{1}{G} \sum_{i=1}^{G} \frac{1}{|o_i|} \sum_{t=1}^{|o_i|} \left[ \frac{\nabla_\theta \pi_\theta\left(o_{i,t}|q, o_{i,<t}\right)}{\pi_{\theta_{old}}(o_{i,t}|q,o_{i,<t})} A_i \right. \right. \\
&\quad \left. \left. + \beta \left( \frac{\pi_{ref}\left(o_{i,t}|q, o_{i,<t}\right) \nabla_\theta \pi_\theta\left(o_{i,t}|q, o_{i,<t}\right)}{\pi_\theta^2\left(o_{i,t}|q, o_{i,<t}\right)} - \frac{\nabla_\theta \pi_\theta\left(o_{i,t}|q, o_{i,<t}\right)}{\pi_\theta\left(o_{i,t}|q, o_{i,<t}\right)} \right) \right] \right\} \\
&= \mathbb{E}_{\left[q \sim P(Q), \{o_i\}_{i=1}^G \sim \pi_{\theta_{old}}(O|q)\right]} \left\{ \frac{1}{G} \sum_{i=1}^{G} \frac{1}{|o_i|} \sum_{t=1}^{|o_i|} \left[ \frac{\pi_\theta\left(o_{i,t}|q, o_{i,<t}\right)}{\pi_{\theta_{old}}\left(o_{i,t}|q, o_{i,<t}\right)} A_i \right. \right. \\
&\quad \left. \left. + \beta \left( \frac{\pi_{ref}\left(o_{i,t}|q, o_{i,<t}\right)}{\pi_\theta\left(o_{i,t}|q, o_{i,<t}\right)} - 1 \right) \right] \frac{\nabla_\theta \pi_\theta\left(o_{i,t}|q, o_{i,<t}\right)}{\pi_\theta\left(o_{i,t}|q, o_{i,<t}\right)} \right\} \\
&= \mathbb{E}_{\left[q \sim P(Q), \{o_i\}_{i=1}^G \sim \pi_{\theta_{old}}(O|q)\right]} \left\{ \frac{1}{G} \sum_{i=1}^{G} \frac{1}{|o_i|} \sum_{t=1}^{|o_i|} \left[ \underbrace{\frac{\pi_\theta\left(o_{i,t}|q, o_{i,<t}\right)}{\pi_{\theta_{old}}\left(o_{i,t}|q, o_{i,<t}\right)} A_i}_{\text{Advantage-weighted probability ratio}} \right. \right. \\
&\quad \left. \left. + \underbrace{\beta \left( \frac{\pi_{ref}\left(o_{i,t}|q, o_{i,<t}\right)}{\pi_\theta\left(o_{i,t}|q, o_{i,<t}\right)} - 1 \right)}_{\text{KL divergence constraint}} \right] \underbrace{\nabla_\theta \log \pi_\theta\left(o_{i,t}|q, o_{i,<t}\right)}_{\text{Policy model gradient}} \right\}.
\end{aligned}
\tag{5}
$$

We analyze the derivative components stressed in Eq. (5). (1) *Advantage-weighted probability ratio term* $\frac{\pi_\theta(o_{i,t}|q,o_{i,<t})}{\pi_{\theta_{old}}(o_{i,t}|q,o_{i,<t})} A_i$ directly ties the contribution of each completion to its advantage. This term incentivizes the policy to prioritize actions with higher rewards, as the advantage function quantifies how much a given action improves expected returns relative to the baseline. By amplifying high-advantage completions and suppressing low-advantage ones, this term guides policy optimization toward reward-aligned reasoning patterns. (2) *KL divergence constraint term* $\beta \left( \frac{\pi_{ref}(o_{i,t}|q,o_{i,<t})}{\pi_\theta(o_{i,t}|q,o_{i,<t})} - 1 \right)$ enforces stability by penalizing deviations from the reference model $\pi_{ref}$. However, this constraint is not inherently designed to shape the policy's reasoning patterns but rather ensures smooth updates during training. (3) *Policy model gradient term* $\nabla_\theta \log \pi_\theta\left(o_{i,t}|q, o_{i,<t}\right)$ represents the gradient of the log-probability of the policy's predicted action with respect to the model parameters $\theta$.

Recent work by Hu *et al.* [10] demonstrates that removing the KL divergence constraint does not impair the trained model's reasoning ability, as the policy's core reasoning patterns are primarily driven by the reward-aligned advantage term. Motivated by this insight, we approximate the policy objective's derivative as:

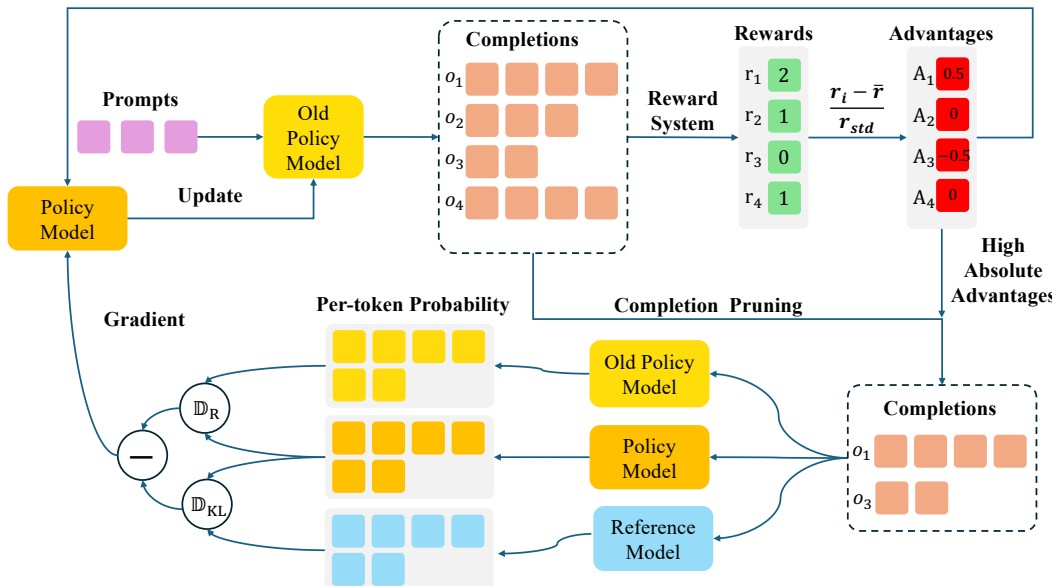

Figure 2: Overview of Completion Pruning Policy Optimization (CPPO). After obtaining the advantages, only completions with high absolute advantage are retained for the forward of the policy model, reference model, and old policy model. $\mathbb{D}_R$ is the probability ratio of the policy to old policy models.

$$\nabla_\theta J_{GRPO}(\theta) \approx \mathbb{E}_{\left[q \sim P(Q), \{o_i\}_{i=1}^G \sim \pi_{\theta_{old}}(O|q)\right]}$$
$$\left\{ \frac{1}{G} \sum_{i=1}^G \frac{1}{|o_i|} \sum_{t=1}^{|o_i|} \left[ \underbrace{\frac{\pi_\theta(o_{i,t}|q, o_{i,<t})}{\pi_{\theta_{old}}(o_{i,t}|q, o_{i,<t})}}_{\substack{\text{Probability ratio} \\ \text{(Post-forward)}}} \cdot \underbrace{A_i}_{\substack{\text{Advantage} \\ \text{(Prior-forward)}}} \right] \underbrace{\nabla_\theta \log \pi_\theta(o_{i,t}|q, o_{i,<t})}_{\substack{\text{Policy model gradient} \\ \text{(Post-forward)}}} \right\}, \quad (6)$$

focusing on the reward-driven learning signal while decoupling the KL regularization constraint.

To better understand this formulation, we decompose the advantage-weighted probability ratio term into the *Probability ratio term* and the *Advantage term*. For a completion that significantly contributes to the policy update, all the new three components in Eq. (6) must be non-negligible. A near-zero or zero value in any of these components would render the overall contribution minimal or nonexistent.

From a computational timing perspective, these components can be categorized as: (1) The probability ratio and policy model gradient are post-forward information, meaning they can only be computed after the policy's forward computation. (2) The advantage term, however, represents prior-forward information that can be calculated before the policy's forward computation.

Given our objective to accelerate GRPO training, we focus on leveraging this prior-forward information. By evaluating the advantage term before the forward computation, we can make an informed decision about whether to process a completion through the policy model. Specifically, if the absolute value of the advantage for a completion is so small or insignificant that it can be practically treated as if it were zero without causing any significant difference in the outcome, we prune that completion from the batch. This selective processing ensures that only completions with high absolute advantage proceed to the forward computation and gradient update stages.

### 3.3 Completions Pruning Policy Optimization

As shown in Figure 2, we propose the Completions Pruning Policy Optimization (CPPO) algorithm to accelerate the training process of group relative policy optimization. Compared to GRPO's optimization objective in Eq. (1), our CPPO introduces a selective condition that only includes completions exhibiting a sufficiently high advantage. The CPPO objective is formulated as follows:

$$\mathcal{J}_{CPPO}(\theta) = \mathbb{E}_{q \sim P(Q), \{o_i\}_{i=1}^{G} \sim \pi_{\theta_{old}}(o|q)} \left\{ \frac{1}{G} \sum_{i=1}^{G} \frac{1}{|o_i|} \sum_{t=1}^{|o_i|} \left\{ \min \left[ \frac{\pi_\theta(o_{i,t}|q, o_{i,<t})}{\pi_{\theta_{old}}(o_{i,t}|q, o_{i,<t})} A_i, \right. \right. \right.$$
$$\left. \left. \left. \text{clip}\left(\frac{\pi_\theta(o_{i,t}|q, o_{i,<t})}{\pi_{\theta_{old}}(o_{i,t}|q, o_{i,<t})}, 1-\epsilon, 1+\epsilon\right) A_i \right] - \beta \mathbb{D}_{KL}\left[\pi_\theta || \pi_{ref}\right] \right\} \right\}, \quad (7)$$
$$s.t. \quad |A_i| \geq \gamma,$$

where $\gamma$ is a predefined threshold that ensures only completions with an absolute advantage above $\gamma$ are retained in the gradient update. It should be noted that when the ratio $\frac{\pi_\theta(o_{i,t}|q, o_{i,<t})}{\pi_{\theta_{old}}(o_{i,t}|q, o_{i,<t})} < 1 - \epsilon$ and $A_i < 0$, or, $\frac{\pi_\theta(o_{i,t}|q, o_{i,<t})}{\pi_{\theta_{old}}(o_{i,t}|q, o_{i,<t})} > 1 + \epsilon$ and $A_i > 0$, the clip function is activated. This action effectively nullifies the policy model gradient term in Eq. (6), equivalent to pruning all completions.

**Unifying Single-/Multiple-GPU(s) Settings**. In a multi-GPUs training scenario, we observe that the number of completions with significant advantages varies across devices. In such cases, the overall training efficiency is bottlenecked by the device processing the largest number of completions—a phenomenon referred to as the bucket effect. To mitigate this, for each GPU, we retain only the $k$ completions with the largest absolute advantage for each question, where

$$k = \lfloor G \times (1 - P) \rfloor, \quad (8)$$

where $P \in (0, 1]$ denoting the pruning rate. The modified CPPO under this strategy is:

$$\mathcal{J}_{CPPO}(\theta) = \mathbb{E}_{q \sim P(Q), \{o_i\}_{i=1}^{G} \sim \pi_{\theta_{old}}(o|q)} \left\{ \frac{1}{k} \sum_{i \in \mathcal{I}} \frac{1}{|o_i|} \sum_{t=1}^{|o_i|} \left\{ \min \left[ \frac{\pi_\theta(o_{i,t}|q, o_{i,<t})}{\pi_{\theta_{old}}(o_{i,t}|q, o_{i,<t})} A_i, \right. \right. \right.$$
$$\left. \left. \left. \text{clip}\left(\frac{\pi_\theta(o_{i,t}|q, o_{i,<t})}{\pi_{\theta_{old}}(o_{i,t}|q, o_{i,<t})}, 1-\epsilon, 1+\epsilon\right) A_i \right] - \beta \mathbb{D}_{KL}\left[\pi_\theta || \pi_{ref}\right] \right\} \right\}, \quad (9)$$

where the summation is taken only over the index set $\mathcal{I}$ corresponding to the $k$ completions with the highest absolute advantage values, *i.e.*,

$$\mathcal{I} = \{i \in \{1, ..., G\} \mid |A_i| \text{ is among the top } k \text{ values}\}. \quad (10)$$

In Sec. 4.2.2, we analyze that completions with high absolute advantage values, either having a correct format and correct answer or an incorrect format and incorrect answer, provide the clearest training signals. Partial correct completions with small absolute advantages contribute minimally or may mislead the policy model. Removing these completions from the training process can enhance training efficiency without compromising model performance.

The key distinction between CPPO and GRPO is that CPPO does not use all completions for the forward computation of the policy model, reference model, and old policy model. Instead, by retaining only those with high absolute advantages for the gradient update, CPPO significantly reduces the computational overhead during forward passes, thereby accelerating the training process.

### 3.4 Parallel Processing through Dynamic Completion Allocation

In this section, we introduce a novel dynamic completion allocation strategy to further optimize the training efficiency of CPPO. Conventional approaches, such as those employed in GRPO, face inherent limitations due to GPU memory constraints. Specifically, a single device can process a maximum of $B$ questions per batch, with each question generating $G$ candidate completions. After pruning, the total number of retained completions per device reduces to $B \times k$, resulting in suboptimal GPU utilization and underleveraged parallel computing capabilities.

To address this inefficiency, we dynamically allocate pruned completions from additional questions into the device's processing pipeline, as illustrated in Figure 3. This strategy ensures that each device operates at full capacity by continuously populating its memory with high-quality completions derived from both the original and newly introduced questions. Critically, all newly incorporated completions undergo the same rigorous pruning process to maintain consistency.

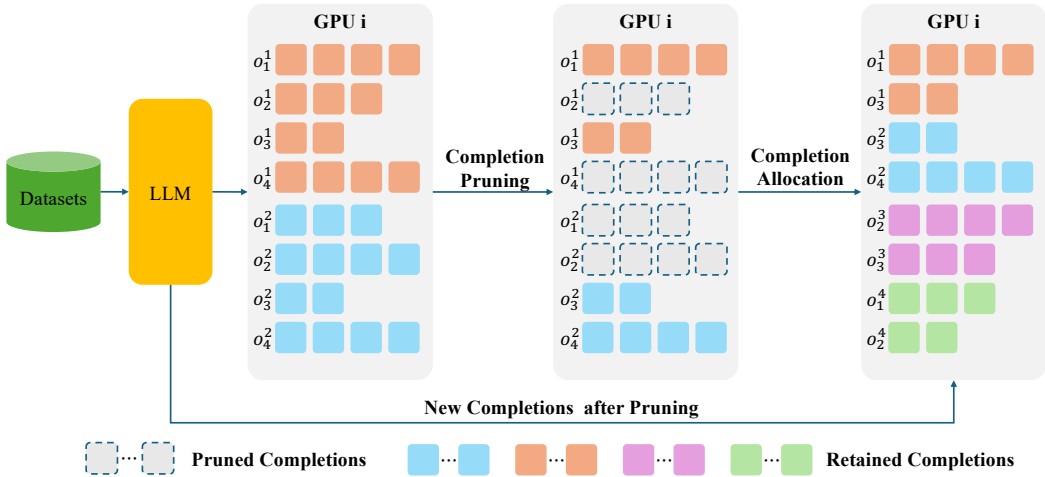

Figure 3: Illustration of dynamic completion allocation for parallel processing. After pruning completions, completion allocation incorporates important completions from new questions. The $o_m^n$ represents the $m$-th completion of the $n$-th question.

The benefits of this approach are twofold. First, it maximizes GPU utilization by fully exploiting the device's parallel computing potential. Second, it enables each device to process a larger number of questions per batch, thereby reducing the total number of training steps required to achieve convergence. This dual optimization boosts training efficiency while maintaining training quality. The CPPO algorithm can be found in Appendix B.

## 4 Experiments

### 4.1 Experimental Settings

**Training Details**. We implement CPPO on the Open R1 [5] and verl [22] frameworks, utilizing the vLLM inference library [14] for efficient completion generation. Qwen2.5-1.5B-Instruct and Qwen2.5-7B-Instruct are trained on two and four GPUs (each with 80GB memory), respectively. We set $\epsilon = 0.2$ and $\beta = 0.04$ in Eq. (7), batch size to 16, number of epochs to 1, and learning rate to $1 \times 10^{-6}$. The policy model temperature is 1, group size is 16, and the maximum completion length is 1024. The prompt templates for CPPO can be found in Appendix C.

**Evaluation Details**. We evaluate the performance on multiple benchmarks with different difficulties, including Math [8], AIME2024 [18], AMC2023 [17], and GSM8K [4]. We use vLLM [14] to accelerate the evaluation process. The evaluation batch size is set to 10. We use greedy decoding to generate completions for Math and GSM8K. For AIME2024 and AMC2023, we set the temperature as $0.6$ and use $4$ completions for each question. We use Pass@1 accuracy as the evaluation metric.

### 4.2 Main Results

We evaluate CPPO by training models of different scales on GSM8K [4] and MATH [8]. GSM8K contains 8.5K grade-school math problems, while MATH includes 7.5K competition-level problems. For the relatively simpler GSM8K dataset, we use Qwen2.5-1.5B-Instruct; for the more challenging MATH dataset, we use Qwen2.5-7B-Instruct. Each model is evaluated on the corresponding test subset. To further assess out-of-distribution reasoning ability, we test Qwen2.5-7B-Instruct on AMC2023 [17] and AIME2024 [18], as these benchmarks are too difficult for Qwen2.5-1.5B-Instruct. Additional results on larger models and different backbones are provided in Appendix D, E, and F. Analyses of stability, convergence, and case studies are presented in Appendix H and I.

#### 4.2.1 Performance Comparison

**Training on GSM8K**. As shown in Table 1, CPPO demonstrates clear advantages over GRPO in both accuracy and acceleration ratio. Notably, CPPO achieves comparable or even higher accuracy

Table 1: Comparison between GRPO and CPPO on GSM8K test subset. We train Qwen2.5-1.5B-Instruct on the GSM8K training subset three times independently to calculate the mean and standard deviation, and the number of retained completions after pruning is denoted by $k = \lfloor G \times (1 - P) \rfloor$.

| Method | Group Size ($G$) | Pruning Rate ($P$) | $k$ | Accuracy (%) | Training Time (s) | Accelerate Ratio |
|---|---|---|---|---|---|---|
| Qwen2.5-1.5B-Instruct | - | - | - | 55.72 | - | - |
| GRPO | 16 | 0.00% | 16 | $77.38 \pm 0.28$ | $23500.33 \pm 130.49$ | 1.00 |
| CPPO | 16 | 50.00% | 8 | $78.15 \pm 0.37$ | $12862.33 \pm 78.68$ | $1.83 \pm 0.01$ |
| CPPO | 16 | 75.00% | 4 | $78.76 \pm 0.25$ | $7436.00 \pm 232.98$ | $3.16 \pm 0.08$ |
| CPPO | 16 | 87.50% | 2 | $80.01 \pm 0.38$ | $4516.33 \pm 237.46$ | $5.22 \pm 0.31$ |
| CPPO | 16 | 93.75% | 1 | $78.99 \pm 1.01$ | $2946.00 \pm 94.44$ | $7.98 \pm 0.23$ |

Table 2: Comparison of GRPO and CPPO on the MATH test subset, as well as on out-of-distribution benchmarks AMC 2023 and AIME 2024. We train Qwen2.5-7B-Instruct on the MATH training dataset three times independently to calculate the mean and standard deviation, and the number of retained completions after pruning is denoted by $k = \lfloor G \times (1 - P) \rfloor$.

| Method | Group Size ($G$) | Pruning Rate ($P$) | $k$ | Accuracy (%) | Training Time (s) | Accelerate Ratio | AMC 2023 | AIME 2024 |
|---|---|---|---|---|---|---|---|---|
| Qwen2.5-7B-Instruct | - | - | - | 55.20 | - | - | 25.62 | 5.00 |
| GRPO | 16 | 0.00% | 16 | $75.26 \pm 0.09$ | $33795.00 \pm 80.18$ | 1.00 | 46.88 | 5.83 |
| CPPO | 16 | 50.00% | 8 | $76.01 \pm 1.03$ | $20129.00 \pm 298.95$ | $1.68 \pm 0.02$ | 53.12 | 10.00 |
| CPPO | 16 | 75.00% | 4 | $76.55 \pm 0.83$ | $13067.00 \pm 81.26$ | $2.59 \pm 0.02$ | 49.38 | 6.67 |
| CPPO | 16 | 87.50% | 2 | $75.95 \pm 0.55$ | $9722.00 \pm 78.87$ | $3.48 \pm 0.03$ | 46.25 | 8.33 |
| CPPO | 16 | 93.75% | 1 | $74.65 \pm 1.31$ | $7608.00 \pm 542.36$ | $4.46 \pm 0.29$ | 45.00 | 5.83 |

than GRPO across various pruning rates. At a pruning rate of 87.50%, CPPO attains an accuracy of 80.01%, surpassing GRPO's 77.38% by 2.63%.

For efficiency, CPPO greatly accelerates training. At a pruning rate of 93.75%, it achieves an acceleration ratio of $7.98\times$. The speedup stems from the completions pruning and the completions allocation. Completions pruning reduces computational overhead by discarding less important completions, while the completions allocation strategy maximizes the use of freed memory and leverages the GPU's parallel processing capabilities. As a result, CPPO processes more questions per batch and reduces the total number of training steps required. These results demonstrate that CPPO not only maintains or improves accuracy but also significantly enhances training efficiency, making it a practical and effective solution for large-scale reasoning model training.

**Training on MATH**. In Table 2, CPPO can well scale to larger models, achieving up to $3.48\times$ acceleration on the MATH without sacrificing accuracy. For instance, at a pruning rate of 87.5%, CPPO attains 75.95% accuracy, outperforming GRPO (75.26%) while cutting training time by $3.48\times$.

Furthermore, evaluation on the AMC2023 and AIME2024 benchmarks confirms that CPPO, despite training only on high absolute advantage completions, preserves the model's generalization ability on out-of-distribution tasks. Thus, CPPO not only matches or even surpasses GRPO in enhancing reasoning capabilities but also well reduces training time, making it a more efficient alternative.

#### 4.2.2 An In-depth Analysis of CPPO's Higher Accuracy

Results on Sec. 4.2.1 indicate that CPPO sometimes achieves better performance at higher pruning rates on the GSM8K and MATH datasets. For example, CPPO with a 75% pruning rate achieves 78.76% accuracy on GSM8K and 76.55% accuracy on MATH, compared to 78.15% and 76.01% accuracy with a 50% pruning rate, respectively. To rule out the possibility that this improvement is merely due to the increased number of questions processed per training step, which is enabled by the completion allocation strategy, we compare CPPO with GRPO under the same number of questions per training step, as shown in Figure 4.

The key difference is that CPPO first generates a group of completions and retains only the top $k$ completions for gradient update, whereas GRPO directly generates $k$ completions for update. Despite this, CPPO consistently outperforms GRPO, demonstrating that its accuracy gains stem

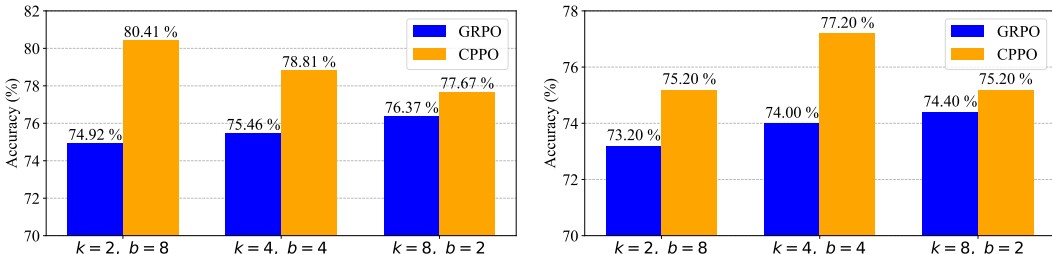

Figure 4: Evaluation accuracy comparison. **Left**: Qwen2.5-1.5b-Instruct on GSM8K test subset. **Right**: Qwen2.5-7b-Instruct on MATH test subset. Here, $k$ denotes the retained completion quantity by CPPO (or generated by GRPO), and $b$ represents questions per training step.

not from processing more questions per step but from the higher quality of retained completions. The quality of completions plays a crucial role in training. CPPO selectively retains high absolute advantage completions from a larger pool, whereas GRPO updates the model with directly generated completions, which may vary in quality. This aligns with our completion contribution analysis in Sec. 3.2, which highlights that completions with high absolute advantages contribute more effectively to training. In GRPO with a group size of 16, both high and low absolute advantage completions are used for training. In contrast, CPPO with a pruning rate of $87.50\%$ trains exclusively on high-advantage completions—yet still achieves superior performance.

To better understand this, we categorize completions into four types: **(1) Correct format and correct answer** — Guides the model to generate accurate completions. **(2) Incorrect format and incorrect answer** — Helps the model avoid incorrect completions. **(3) Correct format and incorrect answer** — May mislead the model into generating partially correct responses. **(4) Incorrect format but correct answer** — Similarly, may introduce noise in learning. Specific examples of the four types of completions can be found in Appendix J.

The first two types are high-quality completions that provide clear training signals. The latter two types, however, are low-quality completions, as their small positive advantage values can mislead the model and introduce noise. Unlike GRPO, which trains on all completions indiscriminately, CPPO filters out these low-quality completions through completions pruning, leading to more efficient learning and better overall performance. As more low-quality completions are removed (pruning rate: $0.00\% \to 87.50\%$), performance improves, as shown in Table 1 and Table 2. However, an excessively high pruning rate can also discard high-quality completions, reducing training effectiveness. This is evident in CPPO's performance decline at a 93.75% pruning rate, as shown in Table 1 and Table 2.

### 4.3 Generalizing CPPO to Other Reinforcement Learning Algorithms

CPPO reduces training cost by pruning low-quality completions. Therefore, CPPO can be generalized to other group relative policy optimization based algorithms such as DAPO [30] and Dr.GRPO [16]. As shown in Table 3, CPPO can be combined with DAPO and Dr.GRPO to further improve training speed and accuracy, demonstrating the strong generalizability of CPPO.

Table 3: Comparison between different reinforcement learning algorithms on the GSM8K test subset. We train Qwen2.5-1.5B-Instruct on the GSM8K training subset, and the number of retained completions after pruning is $k = \lfloor G \times (1 - P) \rfloor$. All experiments are conducted on the verl framework, which supports various RL algorithms.

| Method | Group Size ($G$) | Pruning Rate ($P$) | $k$ | Accuracy (%) | Training Time (s) | Accelerate Ratio ($\times$) |
|---|---|---|---|---|---|---|
| Qwen2.5-1.5B-Instruct | - | - | - | 55.19 | - | - |
| GRPO | 16 | 0.00% | 16 | 78.01 | 7741 | 1.00 |
| CPPO | 16 | 50.00% | 8 | 78.92 | 5192 | 1.49 |
| DAPO | 16 | 0.00% | 16 | 78.01 | 3800 | 1.00 |
| DAPO + CPPO | 16 | 50.00% | 8 | 78.01 | 2134 | 1.78 |
| Dr.GRPO | 16 | 0.00% | 16 | 79.45 | 7991 | 1.00 |
| Dr.GRPO + CPPO | 16 | 50.00% | 8 | 80.14 | 5122 | 1.56 |

Table 4: Evaluation of CPPO with different pruning metrics on GSM8K. "Largest"/"Smallest" prune completions with the highest/lowest absolute advantages, while "Largest*"/"Smallest*" use raw advantage values. "Random" denotes random pruning.

| Pruning Meric | Group Size | Pruning Rate | Accuracy |
|---|---|---|---|
| Qwen2.5-1.5B-Instruct | - | - | 55.72% |
| Largest* | 16 | 50.0% | 73.32% |
| Smallest* | 16 | 50.0% | 76.83% |
| Largest | 16 | 50.0% | 74.23% |
| Random | 16 | 50.0% | 76.98% |
| Smallest | 16 | 50.0% | **77.67%** |

Table 5: Ablation study on the key components of CPPO. Experiments are conducted on Math [8] using Qwen2.5-7B-Instruct [29].

| Method | Group Size | Pruning Rate | Accuracy | Time | Accelerate Ratio |
|---|---|---|---|---|---|
| GRPO | 16 | 0.0% | 75.20% | 33902s | $1.00\times$ |
| + Completion Pruning | 16 | 50.0% | 75.80% | 27547s | $1.23\times$ |
| + Completion Allocation | 16 | 50.0% | 75.20% | 20550s | $1.65\times$ |

## 4.4 Ablation Study

**Ablation Study on Pruning Metrics**. The analysis in Sec. 3.2 reveals that a completion's impact on policy model training is tied to the absolute value of its advantage— a higher absolute value provides stronger training signals. Based on this insight, we adopt the absolute advantage value as the pruning metric, removing completions with the lowest absolute advantages. As shown in Table 4, "Smallest" achieves the best performance, while "Largest" performs the worst, with "Random" falling in between. Additionally, "Smallest*" and "Largest*", which prune completions based on raw advantage values rather than absolute values, perform worse than "Smallest", confirming that absolute advantage values are a more effective pruning metric. The results align with our analysis in Sec. 3.2 and further validate the effectiveness of pruning based on absolute advantage values.

**Ablation Study on Key Modules**. As shown in Table 5, by discarding unimportant completions, the completion pruning module improves the training efficiency by $1.23\times$. By fully leveraging the benefits brought by completion pruning and the GPU parallel computing capability, the completion allocation strategy further improves the training efficiency to $1.65\times$.

## 5 Limitations and Future Work

CPPO does not reduce the time required for generating completions. When completion generation dominates the overall training time, the speedup of CPPO may be reduced. However, CPPO can benefit from inference acceleration methods [3, 15], which are orthogonal to it, to improve training efficiency further. Due to the limited GPU resources of the academic community, we only evaluate the effectiveness of CPPO on relatively small-scale models (less than 14B) and math datasets, including Math and GSM8K. In the future, we plan to: (1) evaluate CPPO on larger models and more tasks. (2) optimize the completion generation time to boost training efficiency further.

## 6 Conclusion

In this paper, we proposed Completion Pruning Policy Optimization (CPPO) to enhance the training efficiency of GRPO-based reasoning models. By selectively pruning completions based on their relative advantages with a suitable pruning rate, CPPO reduces computational overhead without compromising model performance. Additionally, our dynamic completion allocation strategy fully leverages the benefits of completion pruning and GPU parallelism, further boosting training speed. The results demonstrate that CPPO achieves up to $7.98\times$ speedup on GSM8K and $3.48\times$ on Math while sometimes preserving or even enhancing the accuracy compared to GRPO. These findings highlight CPPO as a practical solution for optimizing reasoning model training at a lower cost.

## Acknowledgments

This work was supported by the National Science Fund for Distinguished Young Scholars (No.62025603), National Science Fund for Excellent Young Scholars (No. 62222602), the National Natural Science Foundation of China (No. U21B2037, No. U22B2051, No. U23A20383, No. 62176222, No. 62176223, No. 62176226, No. 62072386, No. 62072387, No. 62072389, No. 62002305 and No. 62272401), and the Natural Science Foundation of Fujian Province of China (No. 2021J06003, No.2022J06001).

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

# Appendix

## A  Specific Design of the Reward Function

In this paper, we define the reward components for the GSM8K dataset as follows:

$$R_{\text{format}}(o_i) = \begin{cases} 1, & \text{if } o_i \text{ follows the correct format,} \\ 0, & \text{otherwise.} \end{cases}$$

$$R_{\text{accuracy}}(o_i) = \begin{cases} 2, & \text{if } o_i \text{ directly matchs the correct answer,} \\ 1.5 & \text{if } o_i \text{ matchs the correct answer after regular parsing,} \\ 0, & \text{otherwise.} \end{cases}$$

And the reward components for the Math datasets are defined as follows:

$$R_{\text{format}}(o_i) = \begin{cases} 1, & \text{if } o_i \text{ follows the correct format,} \\ 0, & \text{otherwise.} \end{cases}$$

$$R_{\text{accuracy}}(o_i) = \begin{cases} 2, & \text{if } o_i \text{ directly matchs the correct answer,} \\ 0, & \text{otherwise.} \end{cases}$$

## B  Algorithm

We provide the algorithm for our Completion Pruning Policy Optimization (CPPO) in Algorithm 1. For dynamic completion allocation, we adopt a more efficient implementation. Specifically, in Sec. 3.4 of the main paper, we describe completion allocation after completion pruning to provide a more intuitive explanation of our method. However, in the algorithm and our code, we perform completion allocation before completion pruning. This is because the number of completions to allocate can be pre-computed based on the pruning rate $p$. Thus, we first sample a batch of $b/(1-p)$ questions from the dataset $\mathcal{D}$ and then sample $G$ completions for each question. This modification does not affect the final results but improves the efficiency of our CPPO, as it benefits from parallelization and inference optimization of vLLMs by sharing prefixes [14].

---

**Algorithm 1** Completions Pruning Policy Optimization

**Input** initial model $\pi_{\theta_{\text{init}}}$; datasets $\mathcal{D}$; datasets size $N$; batch size $b$; hyperparameters $\epsilon$, $\beta$, $\mu$; pruning rate $p$; Group size $G$

1: Policy model $\pi_\theta \leftarrow \pi_{\theta_{\text{init}}}$
2: Reference model $\pi_{ref} \leftarrow \pi_\theta$
3: $M \leftarrow (N \times (1-p))/b$
4: **for** step = 1, ..., $M$ **do**
5:     Update the old policy model $\pi_{\theta_{\text{old}}} \leftarrow \pi_\theta$
6:     # Dynamic Completion Allocation According to Pruning Rate $p$
7:     Sample a batch of $b/(1-p)$ questions $\mathcal{D}_b$ from $\mathcal{D}$
8:     Sample $G$ completions $\{o_i\}_{i=1}^G \sim \pi_{\theta_{\text{old}}}(\cdot \mid q)$ for each question $q \in \mathcal{D}_b$
9:     Compute rewards $\{r_i\}_{i=1}^G$ for each sampled completion $o_i$ via Eq. (4) in the main paper
10:    Compute advantages $A_i$ of $o_i$ according to Eq. (3) in the main paper
11:    # Completion Pruning
12:    Retain $k \leftarrow G \times (1-p)$ completions with the highest absolute advantages for each question
13:    **for** CPPO iteration = 1, ..., $\mu$ **do**
14:       Update the policy model $\pi_\theta$ based on the selected completions (Eq. (9))

**Output** $\pi_\theta$

---

## C  Templates of Completion Pruning Policy Optimization

In this section, we provide the templates of our Completion Pruning Policy Optimization for GSM8K [4] and Math [8] datasets. The templates are shown in Table 6 and Table 7.

Table 6: The templates of our CPPO for GSM8K [4] dataset. Question will be replaced by the question in the dataset.

A conversation between User and Assistant. The user asks a question, and the Assistant solves it. The assistant first thinks about the reasoning process in the mind and then provides the user with the answer. The reasoning process and answer are enclosed within <think> </think> and <answer> </answer> tags, respectively, i.e., <think> reasoning process here </think>\n<answer> answer here </answer>. User: Question. Assistant:

Table 7: The templates of our CPPO for Math [8] dataset. Question will be replaced by the question in the dataset.

A conversation between User and Assistant. The user asks a question, and the Assistant solves it. The assistant first thinks about the reasoning process in the mind and then provides the user with the answer. And the answer should be of the following format: "Therefore, the final answer is: \boxed{ANSWER}. I hope it is correct." (without quotes) where ANSWER is just the final number or expression that solves the problem. The reasoning process and answer are enclosed within <think> </think> and <answer> </answer> tags, respectively, i.e., <think> reasoning process here </think>\n<answer> answer here </answer>. User: Question. Assistant:

## D  Experimental Results on Larger Models

We conduct experiments on larger models, specifically Qwen2.5-14B-Instruct, and present the results in Table 8. The results show that our CPPO can also accelerate the training of larger models by up to $2.83\times$ without compromising accuracy, demonstrating the scalability and generalizability of CPPO.

Table 8: Comparison of GRPO and CPPO on the MATH test subset. We train Qwen2.5-14B-Instruct on the MATH training dataset, and the number of retained completions after pruning is denoted by $k = \lfloor G \times (1 - P) \rfloor$.

| Method | Group Size | Pruning Rate | $k$ | Accuracy | Time | Accelerate Ratio |
|---|---|---|---|---|---|---|
| Qwen2.5-14B-Instruct | - | - | - | 67.80% | - | - |
| GRPO | 16 | 0.00% | 16 | 77.60% | 33942s | $1.00\times$ |
| CPPO | 16 | 50.00% | 8 | 79.40% | 21375s | $1.59\times$ |
| CPPO | 16 | 75.00% | 4 | 79.40% | 15385s | $2.21\times$ |
| CPPO | 16 | 87.50% | 2 | 78.00% | 11997s | $2.83\times$ |
| CPPO | 16 | 93.75% | 1 | 76.00% | 11247s | $3.02\times$ |

## E  CPPO Results on GSM8K with Qwen2.5-7B-Instruct

We additionally conduct experiments with Qwen2.5-7B-Instruct on the GSM8K dataset. As shown in Table 9, GRPO improves accuracy by $8.37\%$ over the baseline, which is less than the $21.66\%$ improvement observed for Qwen2.5-1.5B-Instruct in Table 1. This is because GSM8K is relatively easy for the 7B model, which already achieves a high initial accuracy ($83.00\%$), leaving limited room for improvement. Nevertheless, CPPO still delivers up to $4.67\times$ speedup without loss of accuracy, demonstrating its robustness.

## F  Experimental Results on Different LLM Backbones

We conduct experiments on the Llama series models and present the results in Table 10. The results show that CPPO can also accelerate the training of Llama models without compromising accuracy and achieving a significant speedup of up to $3.13\times$. This demonstrates the generalizability of CPPO across different LLM backbones.

Table 9: Comparison of GRPO and CPPO on the GSM8K test subset. We train Qwen2.5-7B-Instruct on the GSM8K training dataset, and the number of retained completions after pruning is denoted by $k = \lfloor G \times (1 - P) \rfloor$.

| Method | Group Size ($G$) | Pruning Rate ($P$) | $k$ | Accuracy (%) | Training Time (s) | Accelerate Ratio ($\times$) |
|---|---|---|---|---|---|---|
| Qwen2.5-7B-Instruct | - | - | - | 83.00 | - | - |
| GRPO | 16 | 0.00% | 16 | 91.37 | 19294 | 1.00 |
| CPPO | 16 | 50.00% | 8 | 91.59 | 12177 | 1.58 |
| CPPO | 16 | 75.00% | 4 | 91.82 | 7037 | 2.74 |
| CPPO | 16 | 87.50% | 2 | 92.04 | 4975 | 3.88 |
| CPPO | 16 | 93.75% | 1 | 92.04 | 4128 | 4.67 |

Table 10: Comparison between GRPO and CPPO on the GSM8K test subset. We train Llama-3.2-1B-Instruct on the GSM8K training subset, and the number of retained completions after pruning is denoted by $k = \lfloor G \times (1 - P) \rfloor$. Experiments are conducted on the verl [22] framework.

| Method | Group Size ($G$) | Pruning Rate ($P$) | $k$ | Accuracy (%) | Training Time (s) | Accelerate Ratio ($\times$) |
|---|---|---|---|---|---|---|
| Llama-3.2-1B-Instruct | - | - | - | 46.55 | - | - |
| GRPO | 16 | 0.00% | 16 | 62.32 | 13487 | 1.00 |
| CPPO | 16 | 50.00% | 8 | 62.55 | 8362 | 1.61 |
| CPPO | 16 | 75.00% | 4 | 62.62 | 4310 | 3.13 |

# G    Comparison with Other Reinforcement Learning Algorithms

As shown in Table 11, we compare our CPPO with other reinforcement learning algorithms using the GSM8K test subset on the verl [22] framework, which supports various reinforcement learning algorithms. CPPO achieves the best accuracy of $78.92\%$ with a training time of 5192s, outperforming REINFORCE++ [9] and PPO [20] without KL divergence.

Table 11: Comparison between different reinforcement learning algorithms on the GSM8K test subset. We train Qwen2.5-1.5B-Instruct on the GSM8K training subset, and the number of retained completions after pruning is $k = \lfloor G \times (1 - P) \rfloor$. All experiments are conducted on the verl framework.

| Method | Group Size ($G$) | Pruning Rate ($P$) | $k$ | Accuracy (%) | Training Time (s) |
|---|---|---|---|---|---|
| Qwen2.5-1.5B-Instruct | - | - | - | 55.19 | - |
| GRPO | 16 | 0.00% | 16 | 78.01 | 7741 |
| REINFORCE++ | 16 | 0.00% | 16 | 78.54 | 9043 |
| PPO w/o KL | 1 | 0.00% | 1 | 74.45 | 5465 |
| CPPO | 16 | 50.00% | 8 | 78.92 | 5192 |

# H    Stability and Convergence

We plot the reward curves in Figure 5 and 6 during training on both GSM8K and MATH datasets. Overall, the reward curves provide evidence that CPPO preserves GRPO's training stability while improving convergence speed. The results show that the reward curves of CPPO do not crash or experience drastic fluctuations, which is crucial for stable training. These results suggest CPPO's robust and stable training. Moreover, the reward curves of CPPO show a clear upward trend, reaching higher reward values more quickly than GRPO. This faster increase in reward values indicates that CPPO converges more rapidly.

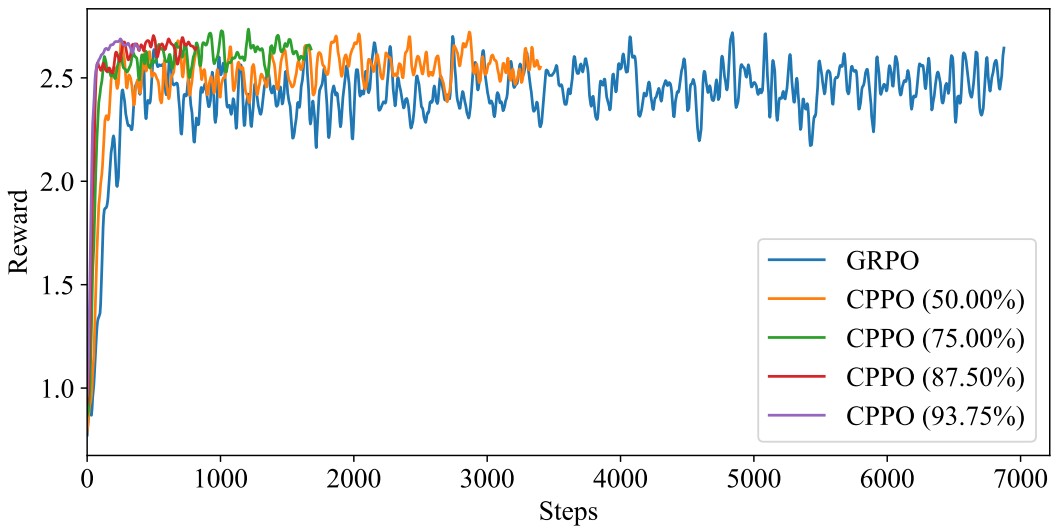

Figure 5: Reward comparison during training (1 epoch) between GRPO and CPPO on GSM8K dataset.

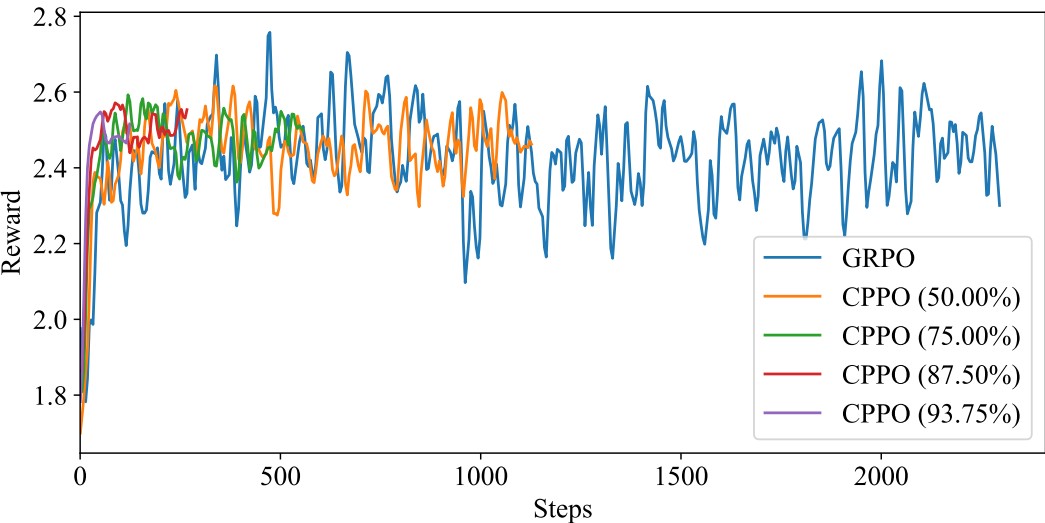

Figure 6: Reward comparison during training (1 epoch) between GRPO and CPPO on the Math dataset.

# I Case Studies

As shown in Tables 12 and 13, we provide case studies to illustrate the differences between completions generated by GRPO and CPPO. We observe that CPPO can maintain or even improve the reasoning ability of the trained model by pruning completions with low absolute advantages during training. Notably, CPPO significantly reduces the training time, achieving up to $7.98\times$ acceleration on GSM8K datasets. Furthermore, in some cases where GRPO fails to provide the correct answer, CPPO with a high pruning rate succeeds, demonstrating its effectiveness in enhancing model performance.

Table 12: The comparisons of completions generated by GRPO and CPPO. The question is sampled from the GSM8K [4] dataset. We use Qwen2.5-1.5B-Instruct [29] model with different training methods to generate completions. The "✓" and "✗" denote a correct and wrong response, respectively.

**Question**:
Ian has a board that is 40 feet long. He decides to make a cut so he can have two pieces. The longer piece is 4 times longer than the shorter piece. How long is the longer piece?

**GRPO Response ✓**:
<think> Let's denote the length of the shorter piece as $L$. According to the problem, the longer piece is 4 times longer than the shorter piece, so the longer piece's length is $4L$. The total length of the board is the sum of the lengths of the two pieces, which is 40 feet. Therefore, we have the equation: $L + 4L = 40$. Simplifying this, we get $5L = 40$. Solving for $L$, we get $L = 8$ feet. Since the longer piece is 4 times longer than the shorter piece, the longer piece's length is $4 \times 8 = 32$ feet. </think>
<answer> 32 </answer>

**CPPO (50.00%) Response ✓**:
<think> Let's denote the length of the shorter piece as $x$. Then the longer piece would be $4x$. The total length of the board is the sum of the lengths of the two pieces, so $x + 4x = 40$. This simplifies to $5x = 40$. Solving for $x$, we get $x = 8$. Therefore, the longer piece is $4x = 4 \times 8 = 32$ feet. </think>
<answer> 32 </answer>

**CPPO (75.00%) Response ✓**:
<think> Let's denote the length of the shorter piece as $x$. Then, the length of the longer piece is $4x$. The total length of the board is the sum of the lengths of the two pieces, so we have the equation: $x + 4x = 40$. This simplifies to $5x = 40$. Solving for $x$, we get $x = 8$. Therefore, the length of the longer piece is $4x = 4 \times 8 = 32$ feet. </think>
<answer> 32 </answer>

**CPPO (87.5%) Response ✓**:
<think> Let's denote the length of the shorter piece as $x$ feet. According to the problem, the longer piece is 4 times longer than the shorter piece, so the longer piece is $4x$ feet. The total length of the board is the sum of the lengths of the two pieces, which is 40 feet. Therefore, we can write the equation: $x + 4x = 40$. Simplifying the left side, we get $5x = 40$. To find $x$, we divide both sides by 5, so $x = 8$. This means the shorter piece is 8 feet long. The longer piece is 4 times the length of the shorter piece, so it is $4 \times 8 = 32$ feet. </think>
<answer> 32 </answer>

**CPPO (93.75%) Response ✓**:
<think> Let's denote the length of the shorter piece as $x$. According to the problem, the longer piece is 4 times longer than the shorter piece, so the length of the longer piece is $4x$. The total length of the board is the sum of the lengths of the two pieces, so we can write the equation: $x + 4x = 40$. Combining like terms, we get $5x = 40$. To find $x$, we divide both sides by 5: $x = 8$. So, the length of the shorter piece is 8 feet. The longer piece is 4 times the length of the shorter piece, so it is $4 \times 8 = 32$ feet. Therefore, the longer piece is 32 feet long. </think>
<answer> 32 </answer>

Table 13: The comparisons of completions generated by GRPO and CPPO. The question is from the GSM8K [4] dataset and generated by Qwen2.5-1.5B-Instruct [29]. The "✓" denotes a correct response while the "✗" indicates a wrong response.

**Question**:
Lee used to be able to run the 400-meter hurdles two seconds faster than Gerald would run the 400-meter hurdles. But Gerald changed his diet, which improved his speed by 10%. If Lee runs the 400-meter hurdles in 38 seconds, how fast can Gerald, with his improved diet, run the 400-meter hurdles, in seconds?

**GRPO Response ✗**:
<think> Let's break down the problem step by step. Lee used to be able to run the 400-meter hurdles two seconds faster than Gerald. This means that if Lee runs in 38 seconds, Gerald would run in 38 + 2 = 40 seconds. Gerald's diet improved his speed by 10%, so we need to calculate his new speed. If Gerald's original speed was $S$ seconds, his improved speed would be $S \times 1.1$. We know that Lee's speed is 2 seconds faster than Gerald's original speed, so we can set up the equation: $38 = S - 2$. Solving for $S$, we get $S = 38 + 2 = 40$ seconds. Now, we can calculate Gerald's improved speed: $40 \times 1.1 = 44$ seconds. </think>
<answer> 44 </answer>

**CPPO (50.00%) Response ✗**:
<think> Let's break down the problem step by step. First, we need to determine how long it used to take Lee to run the 400-meter hurdles. Since Lee used to be able to run it two seconds faster than Gerald, and Lee now takes 38 seconds, we can find out how long it used to take Lee to run the hurdles. Let's denote the time it took Lee to run the hurdles before the diet change as $T$ seconds. Since Lee now takes 38 seconds and he used to be two seconds faster, we have $T - 2 = 38$. Solving this equation, we get $T = 40$ seconds. This means that before the diet change, Lee took 40 seconds to run the 400-meter hurdles. Now, we need to find out how long it took Gerald to run the 400-meter hurdles before the diet change. Since Lee used to be able to run it two seconds faster than Gerald, we have $T_{\text{Gerald}} = 40 - 2 = 38$ seconds. This means that before the diet change, Gerald took 38 seconds to run the 400-meter hurdles. Now, we need to find out how fast Gerald can run the 400-meter hurdles with his improved diet. Since his diet improved his speed by 10%, we need to calculate 110% of his original speed. Let's denote the original speed of Gerald as $S$ meters per second. Since he took 38 seconds to run the 400-meter hurdles before the diet change, we have $S = \frac{400}{38}$ meters per second. With his improved diet, his speed is now $1.1 \times S$. Substituting the value of $S$, we get $1.1 \times \frac{400}{38} = \frac{440}{38}$ meters per second. Simplifying this, we get $\frac{440}{38} = 11.57$ meters per second. </think>
<answer> 11.57 </answer>

**CPPO (75.00%) Response ✗**:
<think> First, we need to determine how long it used to take Lee to run the 400-meter hurdles. Since Lee used to be able to run it two seconds faster than Gerald, and Lee runs it in 38 seconds, we can calculate the time it used to take Gerald to run it. The time it used to take Gerald is $38 + 2 = 40$ seconds. Next, we need to calculate how fast Gerald can run the 400-meter hurdles with his improved diet. His speed improved by 10%, so we need to find 110% of his original time. The original time for Gerald is 40 seconds, so the improved time is $40 \times 1.1 = 44$ seconds. </think>
<answer> 44 </answer>

**CPPO (87.5%) Response ✓**:
<think> Let's break down the problem step by step. First, we need to determine Gerald's original time to run the 400-meter hurdles. We know that Lee is two seconds faster than Gerald, and Lee runs the 400-meter hurdles in 38 seconds. Therefore, Gerald's original time is $38 + 2 = 40$ seconds. Next, we need to calculate Gerald's improved speed after his diet. Gerald's diet improved his speed by 10%. To find his improved time, we need to calculate 10% of 40 seconds and subtract it from 40 seconds. 10% of 40 is $40 \times 0.10 = 4$ seconds. So, Gerald's improved time is $40 - 4 = 36$ seconds. </think>
<answer> 36 </answer>

**CPPO (93.75%) Response ✓**:
<think> Lee used to be able to run the 400-meter hurdles two seconds faster than Gerald would run the 400-meter hurdles. So, if Lee runs in 38 seconds, Gerald used to run in $38 + 2 = 40$ seconds. Gerald's diet improved his speed by 10%. To find out how fast Gerald can run now, we need to calculate 10% of 40 seconds and subtract it from 40 seconds. 10% of 40 seconds is $0.10 \times 40 = 4$ seconds. So, Gerald's new time is $40 - 4 = 36$ seconds. </think>
<answer> 36 </answer>

# J  Illustration of Different Completion Types

We provide examples of different completion types generated by Qwen2.5-1.5B-Instruct [29] in Table 14. These types are as follows: **(1) Correct format and correct answer** — Guides the model to generate accurate and well-structured completions. **(2) Incorrect format and incorrect answer** — Helps the model learn to avoid generating entirely incorrect completions. **(3) Correct format but incorrect answer** — Highlights cases where the format is correct, but the answer is incorrect, potentially misleading the model. **(4) Incorrect format but correct answer** — Demonstrates cases where the answer is correct, but the format is incorrect, which may introduce noise during training.

Table 14:  Illustration of completions from Qwen2.5-1.5B-Instruct [29], using a question from GSM8K [4]. The "✓" ("✗") denotes correct (incorrect) formats or answers.

| |
| --- |
| **Question**: |
| The gauge on a water tank shows that the tank is 1/3 full of water. To fill the tank, 16 gallons of water are added. How many gallons of water does the tank hold when full? |
| **Completion Type 1**: |
| <think> Let's denote the full capacity of the tank by C gallons. The gauge shows that the tank is 1/3 full which means it is 1/3 of C gallons. If we add 16 gallons to the tank, it will be filled completely. So, C + 16 = C * 2/3 (since 1/3 + 1/3 + 1/3 is equal to 1, and 1/f + 1/f + 1/f is equal to 3/f from geometry). Simplifying this equation, we get C = 16 * 3/2 = 24 gallons. </think> 
 <answer> The tank holds 24 gallons when full. </answer> 
 **Format Reward**: ✓ 
 **Accuracy Reward**: ✓ 
 **Advantage**: 2.01 |
| **Completion Type 2**: |
| <think> Let's denote the total capacity of the tank as $C$ gallons. According to the problem, when the tank is 1/3 full, it contains 1/3 $C$ gallons of water. When 16 gallons are added to this, the total becomes full, which means $C = 16 + \frac{1}{3}C$. This equation helps us find the total capacity $C$. </think> 
 **Format Reward**: ✗ 
 **Accuracy Reward**: ✗ 
 **Advantage**: -1.26 |
| **Completion Type 3**: |
| <think>We can set up an equation based on the information in the problem. Let x be the total number of gallons the tank holds when full. </think> 
 <answer> x </answer> 
 **Format Reward**: ✓ 
 **Accuracy Reward**: ✗ 
 **Advantage**: 0.04 |
| **Completion Type 4**: |
| <think> Let's reason through this step by step: </think> 1. We know that the tank was 1/3 full. 2. To find out the total capacity of the tank when it's full, we need to multiply the amount already in the tank by 3 (since 1 full tank = 3/3 full tanks). 3. We also know that adding 16 gallons filled the tank to its full capacity. 4. So, if 16 gallons equals the difference between the full tank and the current amount, we can use this information to find the current amount in gallons. 5. If 16 gallons are added to 4/3 of the total capacity to get the full capacity, then the current amount must be 3/4 of the full capacity. 6. Let's call the total capacity of the tank C (it's a whole number, in gallons). So, the current amount in the tank = (3/4)C 7. Rearranging 3/4 to look like 3C/4, and knowing that (3C/4) = C - (1/3C) = 2/3C, and we know that 2/3C = 16, 8. Therefore 2/3 * C = 16, 9. So C must be 16 * 3/2, 10. Which equals 24 gallons. <answer> The tank holds 24 gallons when full. </answer> 
 **Format Reward**: ✗ 
 **Accuracy Reward**: ✓ 
 **Advantage**: 0.69 |

