# OpenReview forum: "CPPO: Accelerating the Training of Group Relative Policy Optimization-Based Reasoning Models"
_NeurIPS.cc/2025/Conference — NeurIPS 2025 poster_

### Official Review · Reviewer_ZKXt · 2025-07-02

**Clarity:** 2
**Significance:** 3
**Originality:** 3
**Rating:** 5
**Confidence:** 3

**Summary:**

This paper introduces CPPO, a novel RL method to train language models for reasoning tasks. The method builds on top of GRPO, and adjusts it to only use a subset of completions with high absolute advantages, reducing computation required to process a sample. To compare this method against GRPO, Qwen 2.5 (1.5B and 7B) models are trained and evaluated on mathematical reasoning datasets. The results show that CPPO reduces training time and certain configurations of CPPO outperform GRPO in accuracy.

**Questions:**

1.How do you compare in speedup and accuracy against other mentioned RL algorithms (REINFORCE++, PPO without KL divergence)?

2.Why did you choose to use different models for different datasets? Can you provide some data showing it wouldn't be reasonable to do so?

3.What was the stopping criterion for training?

4.High pruning rates with the Qwen 7B model don't result in increase in accuracy across MATH and AMC2023 datasets, yet you claim that "CPPO achieves better performance at high pruning rates". Can you explain this, as these two seem contradictory?

**Ethical Concerns:**

["NO or VERY MINOR ethics concerns only"]

**Final Justification:**

The authors answered reviewers' questions well and added additional evaluations, as requested. The additional results add robustness to the performance evaluation of the method. Authors also improved clarity on multiple subjects raised by reviewers. I suggest to accept this paper.

**Limitations:**

Yes

**Paper Formatting Concerns:**

No concerns

**Quality:**

3

**Strengths And Weaknesses:**

Strengths
1. CPPO is a novel method that tackles a significant problem of training efficiency of GRPO. The authors provide clear motivation and reasons behind their modifications to the algorithm.
2. Authors provide extensive algorithm description and code to replicate results.
3. Comparison is done with relevant benchmarks in the mathematical reasoning space.
4. The train time speedup, while retaining accuracy, is impressive, positioning CPPO as a viable alternative to GRPO.

Weaknesses
1. Other RL algorithms, including the ones mentioned in the related work section, are not compared against. Authors mention these methods can have substantial computational cost, however it is unclear how much CPPO improves the speed and accuracy  of training compared to this broader space of RL algorithms.
2. The chosen metric is pass@1, which can be highly variable, but the variance of this metric is not captured. Additionally, evaluated models differ between datasets. Combined, this undermines the authenticity of the results. Ideally authors would evaluate the algorithm across the same models and capture the variance of results.
3. The authors claim "CPPO reduces computational overhead without compromising model performance", however high pruning rate configurations of the CPPO-trained 7B Qwen model have lower accuracies on MATH and AMC compared to GRPO. The claim should be amended.
4. The stopping criterion for training is not discussed in the 4.1 section.
5. In figure 4 it is not clear how this demonstrates that accuracy stems from higher quality completions.

---

> ### Author Rebuttal · Authors · 2025-07-29
>
> We sincerely appreciate your encouraging feedback on our work, including comments like "CPPO is a novel method that tackles a significant problem of training efficiency of GRPO", "clear motivation", and "The train time speedup, while retaining accuracy, is impressive, positioning CPPO as a viable alternative to GRPO."  Please see our point-by-point responses to your comments below.
>
>
> **Q1: How do you compare in speedup and accuracy against other mentioned RL algorithms (REINFORCE++, PPO without KL divergence)?**
>
> **A1:** As shown in Table 1, we compare our CPPO with other reinforcement learning algorithms using the GSM8K test subset on the verl[1] framework, which supports various RL algorithms. CPPO achieves the best accuracy of 78.92% with a training time of 5192s, outperforming REINFORCE++ and PPO without KL divergence. We will include these results in the camera-ready version.
>
>
> Table 1: Comparison between different reinforcement learning algorithms on the GSM8K test subset. We train Qwen2.5-1.5B-Instruct on the GSM8K training subset, and the number of retained completions after pruning is
> $k = \left\lfloor G \times (1 - P) \right\rfloor$. All experiments are conducted on the verl framework.
> |Method|Group Size ($G$)|Pruning Rate ($P$)|$k$|Accuracy (\%) |Training Time (s)|
> |-|-|-|-|-|-|
> |Qwen2.5-1.5B-Instruct|-|-|-|55.19|-|
> |GRPO|16|0.00%|16|78.01|7741|
> |REINFORCE++|16|0.00%|16|78.54|9043|
> |PPO w/o KL|1|0.00%|1|74.45|5465|
> |CPPO|16|50.00%|8|78.92|5192|
>
>
> [1] Sheng et al. HybridFlow: A Flexible and Efficient RLHF Framework
>
> **Q2:  Ideally, authors would evaluate the algorithm across the same models and capture the variance of results.**
>
> **A2:**  As requested, for more rigid evaluation, we choose to conduct multiple independent runs for the Qwen2.5-1.5B-Instruct and Qwen2.5-7B-Instruct models and report the results in Tables 2 and 3. The results demonstrate the stability  and robustness of our CPPO.
>
> Table 2: Comparison between GRPO and CPPO on the GSM8K test subset. We independently train Qwen2.5-1.5B-Instruct on the GSM8K training subset three times to calculate the mean and standard deviation, and the number of retained completions after pruning is $k = \left\lfloor G \times (1 - P) \right\rfloor$.
>
> |Method|Group Size ($G$)|Pruning Rate ($P$)|$k$|Accuracy (\%)|Training Time (s)|Accelerate Ratio|
> |-|-|-|-|-|-|-|
> |Qwen2.5-1.5B-Instruct|-|-|-|55.72|-|-|
> |GRPO|16|0.00\%|16|77.38 $\pm$ 0.28|23500.33 $\pm$ 130.49|1.00|
> |CPPO|16|50.00\%|8|78.15 $\pm$ 0.37|12862.33 $\pm$ 78.68|1.83 $\pm$ 0.01|
> |CPPO|16|75.00\%|4|78.76 $\pm$ 0.25|7436.00 $\pm$ 232.98|3.16 $\pm$ 0.08|
> |CPPO|16|87.50\%|2|80.01 $\pm$ 0.38|4516.33 $\pm$ 237.46|5.22 $\pm$ 0.31|
> |CPPO|16|93.75\%|1|78.99 $\pm$ 1.01|2946.00 $\pm$ 94.44|7.98 $\pm$ 0.23|
>
> Table 3: Comparison of GRPO and CPPO on the MATH test subset. We independently train Qwen2.5-7B-Instruct on the MATH training dataset three times to calculate the mean and standard deviation, and the number of retained completions after pruning is denoted by $k = \left\lfloor G \times (1 - P) \right\rfloor$.
>
> |Method|Group Size ($G$)|Pruning Rate ($P$)|$k$|Accuracy (\%)|Training Time (s)|Accelerate Ratio|
> |-|-|-|-|-|-|-|
> |Qwen2.5-7B-Instruct|-|-|-|55.20 |-|-|
> |GRPO|16|0.00\%|16|75.26 $\pm$ 0.09|33795.00 $\pm$ 80.18|1.00|
> |CPPO|16|50.00\%|8|76.01 $\pm$ 1.03|20129.00 $\pm$ 298.95|1.68 $\pm$ 0.02|
> |CPPO|16|75.00\%|4|76.55 $\pm$ 0.83|13067.00 $\pm$ 81.26|2.59 $\pm$ 0.02|
> |CPPO|16|87.50\%|2|75.95 $\pm$ 0.55|9722.00 $\pm$ 78.87|3.48 $\pm$ 0.03|
> |CPPO|16|93.75\%|1|74.65 $\pm$ 1.31|7608.00 $\pm$ 542.36|4.46 $\pm$ 0.29|
>
>
> **Q3: The authors claim "CPPO reduces computational overhead without compromising model performance."; However, high pruning rate configurations of the CPPO-trained 7B Qwen model have lower accuracies on MATH and AMC compared to GRPO. The claim should be amended.**
>
> **A3:** Thank you for pointing this out. We will revise the claim to "CPPO reduces computational overhead while maintaining comparable model performance at an appropriate pruning rate. An excessively high pruning rate may lower the model performance."
>
>
> **Q4: The stopping criterion for training is not discussed in the 4.1 section.**
>
> **A4:** In our experimental settings, we use the same stopping criterion for all methods for fair comparison. Specifically, we set the training epoch to 1, and the training will stop after the model has been trained on the entire training set once. We will add this to Section 4.1 of the main paper.
>
> **Q5: In Figure 4, it is not clear how this demonstrates that accuracy stems from higher quality completions.**
>
> **A5:** In Figure 4, we compare the accuracy of GRPO and CPPO with different question numbers $b$ and retained completion numbers $k$ per training step. The key difference between CPPO and GRPO is that CPPO first generates a group of $G$ ($G > k$) completions and then retains only the top $k$ completions per training step, whereas GRPO directly generates $k$ completions per step. As a result, the completions retained by CPPO are of higher quality, since they are selected from a larger pool based on their absolute advantages. In contrast, the completions retained by GRPO are generated without selection and are more likely to include low-quality completions. The results in Figure 4 show that CPPO achieves higher accuracy than GRPO with all different settings of $b$ and $k$, demonstrating that the improved accuracy is due to the higher quality of completions retained by CPPO.
>
>
> **Q6: Why did you choose to use different models for different datasets? Can you provide some data showing it wouldn't be reasonable to do so?**
>
> **A6:** We choose different models for different datasets for two main reasons: (1) Demonstrating generalizability: Using various models across datasets shows that CPPO works for both small and large models, and on different datasets, highlighting its robustness. (2) Reasonable experimental settings: GSM8K contains 8.5K grade-school math problems, which are relatively simple for larger models like Qwen2.5-7B-Instruct (as shown in Table 3, its GSM8K accuracy is already 83%). MATH, on the other hand, includes 7.5K competition-level problems that require a stronger reasoning ability. Therefore, we use Qwen2.5-1.5B-Instruct for GSM8K and Qwen2.5-7B-Instruct for MATH to ensure the experiments are meaningful and challenging for each model.
>
> Table 3: The initial reasoning capabilities of different models.
>
> |Model|GSM8K Accuracy|MATH Accuracy|
> |-|-|-|
> |Qwen2.5-1.5B-Instruct|55.72%|48.40%|
> |Qwen2.5-7B-Instruct|83.00%|55.20%|
>
>
>
> **Q7: High pruning rates with the Qwen 7B model don't result in an increase in accuracy across MATH and AMC2023 datasets, yet you claim that "CPPO achieves better performance at high pruning rates". Can you explain this, as these two seem contradictory?**
>
>
> **A7:** Thank you for highlighting this issue. We will revise our claim to: "CPPO sometimes achieves better performance at higher pruning rates on the GSM8K and MATH datasets. For example, CPPO with a 75% pruning rate achieves 78.81% accuracy on GSM8K and 77.20% accuracy on MATH, compared to 77.67% and 75.20% accuracy with a 50% pruning rate, respectively." We will also clarify that CPPO with excessively high pruning rates, such as 93.75%, may influence the model performance negatively on some datasets, such as MATH and AMC2023.
>
>
> Thank you again for spending time reviewing our work and providing valuable feedback. We hope our responses address your concerns. If you have any further questions or suggestions, please feel free to let us know.

---

> > ### Comment · Reviewer_ZKXt · 2025-08-02
> >
> > Thanks to the authors for detailed answers to all of the reviewers' questions. Given the one-week rebuttal period and limited computational resources, the authors added a resonable amount of additional evaluations, as requested by multiple reviewers. The additional results resolve some doubts about the performance of the method and reaffirm my original score (Accept). However, I'm not fully satisfied by the answer to Q6 - it would help to at least cite some other evaluations or show that GRPO doesn't improve performance of the 7B model on GSM8K. This would support the claim that this dataset is relatively easy for the 7B model and show that there's no room for improvement.

---

> > > ### Author Response · Authors · 2025-08-04
> > >
> > > Thank you for your constructive suggestions. To better answer Q6, we additionally conducted experiments with Qwen2.5-7B-Instruct on the GSM8K dataset. As shown in Table 1, GRPO improves over the baseline by 8.37% on Qwen2.5-7B-Instruct, which is less than the 21.33% improvement observed on Qwen2.5-1.5B-Instruct. This indicates that GSM8K is a relatively easy dataset for the 7B model, as its initial accuracy is already high (83.00%), leaving limited room for further improvement. Nevertheless, in this setting, CPPO still achieves up to 4.67x speedup without compromising accuracy, demonstrating the robustness of CPPO. We will include these results and this discussion in the final paper. Thank you again for your valuable feedback, which has helped us improve our work.
> > >
> > >
> > >
> > > Table 1: Comparison of GRPO and CPPO on the GSM8K test subset. We train Qwen2.5-7B-Instruct on the GSM8K training dataset, and the number of retained completions after pruning is denoted by $k = \left\lfloor G \times (1 - P) \right\rfloor$.
> > >
> > > |Method|Group Size ($G$)|Pruning Rate ($P$)|$k$|Accuracy (\%)|Training Time (s)|Accelerate Ratio|
> > > |-|-|-|-|-|-|-|
> > > |Qwen2.5-7B-Instruct|-|-|-|83.00|-|-|
> > > |GRPO|16|0.00%|16|91.37|19294|1.00|
> > > |CPPO|16|50.00%|8|91.59|12177|1.58|
> > > |CPPO|16|75.00%|4|91.82|7037|2.74|
> > > |CPPO|16|87.50%|2|92.04|4975|3.88|
> > > |CPPO|16|93.75%|1|92.04|4128|4.67|

---

> > > > ### Comment · Reviewer_ZKXt · 2025-08-05
> > > >
> > > > Thank you for conducting the experiment. This fully answers my question.

---

### Official Review · Reviewer_kqfe · 2025-07-02

**Clarity:** 3
**Significance:** 2
**Originality:** 3
**Rating:** 4
**Confidence:** 4

**Summary:**

This paper proposes Completion Pruning Policy Optimization (CPPO), a method designed to improve the training efficiency of reasoning models built upon Group Relative Policy Optimization (GRPO). The authors theoretically and empirically demonstrate that not all completions contribute equally to policy updates, with their usefulness correlated to relative advantage. CPPO addresses this by pruning completions with low absolute advantages, thereby reducing computational overhead without sacrificing performance. Furthermore, the authors propose a dynamic completion allocation strategy to optimize GPU usage. Experiments on GSM8K and Math datasets show that CPPO achieves substantial speedups (up to 8.32×) while maintaining or improving model accuracy.

**Questions:**

1. Can this algorithm generate to other RL variants?

**Ethical Concerns:**

["NO or VERY MINOR ethics concerns only"]

**Final Justification:**

The acceleration effect is significant. Acceleration of RL is an important direction.

**Limitations:**

Yes

**Quality:**

3

**Strengths And Weaknesses:**

**Strength**:

1.	Accelerating GRPO-style reinforcement learning methods is an important research direction.
2.	The authors present a clear motivation through the experiment in Figure 1, and the other experiments are also thorough and well-executed.
3.	The paper does show impressive speedup ratios, with up to 8.32x on GSM8K and 3.51x on Math, which is a clear practical improvement for GRPO training. This directly addresses a known bottleneck in GRPO, which is computationally expensive due to multiple completions per question.
4.	The core ideas of pruning low-advantage completions and dynamically allocating remaining GPU resources are genuinely novel and seem well-motivated by the analysis of completion contributions. This intelligent filtering of training signals could lead to more efficient learning.

**Weakness**:

1. The experiments are conducted only on the Qwen2.5 series, making it difficult to assess whether the proposed method generalizes to different LLM backbones.
2. As a highly engineering-oriented work, I strongly recommend that the authors release their code, as it would be greatly beneficial to the community.

---

> ### Author Rebuttal · Authors · 2025-07-29
>
> We extend our gratitude to your valuable and encouraging feedback on our work, such as "genuinely novel", "well-motivated", and "well-executed". Please kindly see our responses to your comments below.
>
> **Q1: The experiments are conducted only on the Qwen2.5 series, making it difficult to assess whether the proposed method generalizes to different LLM backbones.**
>
> **A1:** We conduct experiments on the Llama series models and present the results in Table 1. The results show that CPPO can also accelerate the training of Llama models without compromising accuracy and achieving a significant speedup of up to 3.13×. This demonstrates the generalizability of CPPO across different LLM backbones. We will include these results in the camera-ready version.
>
> Table 1: Comparison between GRPO and CPPO on the GSM8K test subset. We train Llama-3.2-1B-Instruct on the GSM8K training subset, and the number of retained completions after pruning is
> $k = \left\lfloor G \times (1 - P) \right\rfloor$. All experiments are conducted on the verl framework, which supports various RL algorithms.
> |Method|Group Size($G$)|Pruning Rate($P$)|$k$|Accuracy|Training Time|Accelerate Ratio|
> |-|-|-|-|-|-|-|
> |Llama-3.2-1B-Instruct|-|-|-|46.55%|-|-|
> |GRPO|16|0.00%|16|62.32%|13487s|1.00×|
> |CPPO|16|50.00%|8|62.55%|8362s|1.61×|
> |CPPO|16|75.00%|4|62.62%|4310s|3.13×|
>
>
> **Q2: As a highly engineering-oriented work, I strongly recommend that the authors release their code, as it would be greatly beneficial to the community.**
>
> **A2:** Thank you for your suggestion. The code is included in the supplementary material. We will also release our code publicly upon acceptance of the paper.
>
> **Q3: Can this algorithm generalize to other RL variants?**
>
> **A3:** CPPO reduces training cost by pruning low-quality completions. Therefore, CPPO can be generalized to other group relative policy optimization based RL algorithms such as DAPO[1] and Dr.GRPO[2]. As shown in Table 2, CPPO can be combined with DAPO and Dr.GRPO to further improve training speed and accuracy, demonstrating the strong generalizability of CPPO.
>
> Table 2: Comparison between different reinforcement learning algorithms on the GSM8K test subset. We train Qwen2.5-1.5B-Instruct on the GSM8K training subset, and the number of retained completions after pruning is
> $k = \left\lfloor G \times (1 - P) \right\rfloor$. All experiments are conducted on the verl framework, which supports various RL algorithms.
> |Method|Group Size ($G$)|Pruning Rate ($P$)|$k$|Accuracy (\%)|Training Time (s)|Accelerate Ratio|
> |-|-|-|-|-|-|-
> |Qwen2.5-1.5B-Instruct|-|-|-|55.19|-|-|
> |GRPO|16|0.00%|16|78.01|7741|1.00|
> |CPPO|16|50.00%|8|78.92|5192|1.49|
> |DAPO|16|0.00%|16|78.01|3800|1.00|
> |DAPO+CPPO|16|50.00%|8|78.01|2134|1.78|
> |Dr.GRPO|16|0.00%|16|79.45|7991|1.00|
> |Dr.GRPO+CPPO|16|50.00%|8|80.14|5122|1.56|
>
> [1] Yu, Qiying, et al. "Dapo: An open-source llm reinforcement learning system at scale." arXiv preprint arXiv:2503.14476 (2025).
> [2] Liu, Zichen, et al. "Understanding r1-zero-like training: A critical perspective." arXiv preprint arXiv:2503.20783 (2025).
>
>
> We hope our responses address your concerns. If you have any further questions, please let us know. Thank you again for your time and valuable feedback.

---

> > ### Comment · Reviewer_kqfe · 2025-08-08
> >
> > Thank you for your response, which partly addressed my concerns. I will keep my score.

---

### Official Review · Reviewer_zbVH · 2025-07-03

**Clarity:** 2
**Significance:** 2
**Originality:** 3
**Rating:** 2
**Confidence:** 4

**Summary:**

This paper introduces Completion Pruning Policy Optimization (CPPO), which prunes low‐advantage completions to reduce the number needed for policy updates and dynamically reallocates freed GPU capacity to new questions, thereby maintaining or improving model accuracy while significantly speeding up training . Experiments on GSM8K and MATH benchmarks demonstrate up to **8.32×** and **3.51×** speedups respectively, with accuracy equal to or exceeding GRPO .

**Questions:**

1. **Threshold Sensitivity**: How sensitive is CPPO’s performance to the choice of γ? Have you evaluated multiple threshold settings to identify an optimal or adaptive strategy?

2. **Integration with Inference Acceleration**: In scenarios where completion generation dominates, have you explored combining CPPO with methods like TokenSkip or speculative sampling to further reduce wall‐clock time?

3. **Generality to Larger Models and Tasks**: Could you extend CPPO’s evaluation to models > 7B parameters and non‐math reasoning benchmarks to validate scalability and robustness?

4. **Robustness of Results**: Have you conducted multiple independent runs to compute error bars or statistical significance measures (e.g., t‐tests) for the reported accuracy and speedup?

**Ethical Concerns:**

["NO or VERY MINOR ethics concerns only"]

**Limitations:**

The paper acknowledges that CPPO **does not reduce completion generation time** and that evaluations are **limited to models under 7B parameters and math benchmarks**, suggesting future work to integrate inference acceleration techniques and broaden the evaluation to larger models and diverse tasks .

**Quality:**

1

**Strengths And Weaknesses:**

## Strengths

* **Theoretical Motivation**
  The authors rigorously analyze completion contributions and show that advantage correlates with training signal, underpinning the pruning strategy (Sec. 3.2, lines 117–123).

* **Dynamic GPU Utilization**
  The dynamic completion allocation mechanism leverages GPU capacity post‐pruning to process additional questions, maximizing throughput (Sec. 3.4, lines 176–183).

* **Empirical Effectiveness**
  CPPO achieves up to **8.32×** acceleration on GSM8K and **3.51×** on MATH without sacrificing—and often improving—accuracy, as shown in Table 1 and Table 2 (lines 219–227; 229–235).

* **Comprehensive Ablations**
  Detailed studies on pruning metrics (“Smallest” vs. “Largest”) and component contributions (pruning vs. allocation) validate design choices and quantify their individual impact (Table 3 and Table 4, lines 266–274).

## Weaknesses

1. **Pruning Threshold Selection (Eq. 7, Line 154)**
   The choice of the absolute‐advantage threshold $\gamma$ is fixed without justification or sensitivity analysis. Could the authors consider an **adaptive threshold** (e.g., via cross‐validation or reinforcement learning) [1] to balance pruning aggressiveness and performance?

2. **Completion Generation Overhead (Limitation Sec. 5)**
   CPPO accelerates forward and backward passes but does not address the time spent generating completions, which can dominate total training time. Integrating inference acceleration methods such as **TokenSkip** [2] or **Eagle** [3] could further reduce end‐to‐end training time.

3. **Narrow Evaluation Scope (Limitation Sec. 5)**
   Experiments are confined to math datasets and models under 7B parameters. Broader validation on larger models (> 7B) and diverse reasoning tasks (e.g., commonsense QA, code generation) would strengthen claims of generality

4. **Lack of Statistical Significance**
   The reported accuracies in Tables 1 and 2 lack error bars or multiple‐seed evaluations, making the result variability unclear. Including **confidence intervals** over repeated runs would bolster confidence in the observed speed/accuracy tradeoffs.


[1] Gu, Jiahua, et al. "Adaptive Threshold-Triggered Heuristic-assisted Deep Reinforcement Learning for Energy-efficient and QoS-guaranteed 5G RAN Slice Migration." GLOBECOM 2024-2024 IEEE Global Communications Conference. IEEE, 2024.

[2] Xia, Heming, et al. "Tokenskip: Controllable chain-of-thought compression in llms." arXiv preprint arXiv:2502.12067 (2025).

[3] Li, Yuhui, et al. "Eagle-2: Faster inference of language models with dynamic draft trees." arXiv preprint arXiv:2406.16858 (2024).

---

> ### Author Rebuttal · Authors · 2025-07-29
>
> Thank you for spending time reviewing our work. Please kindly see our responses to your comments below.
>
> **Q1: Threshold Sensitivity: How sensitive is CPPO’s performance to the choice of $\gamma$? Have you evaluated multiple threshold settings to identify an optimal or adaptive strategy?**
>
> **A1:** Eq. (7) is only an intermediate step in deriving our final policy objective function. Final policy objective function is formally defined in Eq. (9) and Eq. (10) of the main paper. We do not use a fixed threshold $\gamma$ in our final policy objective function. Since the range of advantage values varies across tasks and questions, using a fixed threshold $\gamma$ for completion pruning is unsuitable. During multi-GPU training, we observed that the number of completions exceeding a fixed threshold differs across devices, resulting in overall efficiency being limited by the device with the most completions (the "bucket effect"). Therefore, for each GPU, CPPO retains the $k$ completions with the largest absolute advantages for each question, rather than using a fixed threshold. More details can be found in Section 3.3 of the main paper.
>
> **Q2: Integration with Inference Acceleration: In scenarios where completion generation dominates, have you explored combining CPPO with methods like TokenSkip or speculative sampling to further reduce wall-clock time?**
>
> **A2:** We would like to clarify that inference acceleration methods are orthogonal to our work. Even in scenarios where completion generation dominates, our CPPO can achieve a considerable speedup, such as 2.83x (Table 2 with the 14B model). Inference acceleration is a popular research topic, and we believe that combining CPPO with more advanced inference acceleration methods can lead to further improvements in training speed and efficiency, which will also be our future focus.
>
> **Q3: Generality to Larger Models and Tasks: Could you extend CPPO’s evaluation to models > 7B parameters and non-math reasoning benchmarks to validate scalability and robustness?**
>
> **A3:** As requested, we conduct experiments on larger models, specifically Qwen2.5-14B-Instruct, and present the results in Table 2. The results show that CPPO can accelerate the training of larger models by up to 2.83x without compromising accuracy, demonstrating the scalability and generalizability of our method. While we aim to evaluate our method on even larger models, limited academic computational resources currently restrict us to Qwen2.5-14B-Instruct. Additionally, we evaluate our method on more benchmark datasets in Table 3. Specifically, we train Qwen2.5-7B-Instruct on the MATH training dataset and evaluate the model on non-math QA datasets (GPQA:Diamond) and code generation tasks (LiveCodeBench). The results show that both CPPO and GRPO, when trained on math datasets, achieve improved performance on GPQA. However, due to the large domain gap between math and code generation, performance on code generation tasks is slightly reduced for both methods, with CPPO exhibiting less degradation. This demonstrates that our method can accelerate GRPO training while maintaining the model’s generalization ability. Validating rule-based reinforcement learning algorithms on math tasks is common practice in this field [1][2], so we also choose math tasks to evaluate our method in this paper. Evaluating CPPO on more tasks, such as code generation and multimodal tasks, will be our future work, as these require substantial computational resources and engineering effort.
>
> Table 2: Comparison of GRPO and CPPO on the MATH test subset. We train Qwen2.5-14B-Instruct on the MATH training dataset, and the number of retained completions after pruning is denoted by $k = \left\lfloor G \times (1 - P) \right\rfloor$.
> |Method|Group Size ($G$)|Pruning Rate ($P$)|$k$|Accuracy (\%)|Training Time (s)|Accelerate Ratio|
> |-|-|-|-|-|-|-|
> |Qwen2.5-14B-Instruct|-|-|-|67.80|-|-|
> |GRPO|16|0.00%|16|77.60|33942|1.00|
> |CPPO|16|50.00%|8|79.40|21375|1.59|
> |CPPO|16|75.00%|4|79.40|15385|2.21|
> |CPPO|16|87.50%|2|78.00|11997|2.83|
> |CPPO|16|93.75%|1|76.00|11247|3.02|
>
> Table 3: Comparison between GRPO and CPPO on additional benchmark datasets. We train Qwen2.5-7B-Instruct on the MATH training dataset and evaluate the model on various out-of-distribution benchmarks using lighteval. The number of retained completions after pruning is given by $k = \left\lfloor G \times (1 - P) \right\rfloor$.
>
> |Method|Group Size ($G$)|Pruning Rate ($P$)|$k$|GPQA:Diamond (\%)|LiveCodeBench|
> |-|-|-|-|-|-|
> |Qwen2.5-7B-Instruct|-|-|-|33.46 $\pm$ 2.46|13.99 $\pm$ 1.88|
> |GRPO|16|0.00%|16|33.71 $\pm$ 2.55|13.46 $\pm$ 1.80|
> |CPPO|16|50.00%|8|34.60 $\pm$ 2.70|13.78 $\pm$ 1.83|
>
> [1] Yu, Qiying, et al. "Dapo: An open-source llm reinforcement learning system at scale." arXiv preprint arXiv:2503.14476 (2025).
> [2] Liu, Zichen, et al. "Understanding r1-zero-like training: A critical perspective." arXiv preprint arXiv:2503.20783 (2025).
>
> **Q4: Robustness of Results: Have you conducted multiple independent runs to compute error bars or statistical significance measures (e.g., t-tests) for the reported accuracy and speedup?**
>
> **A4:** GRPO experiments are computationally expensive and time-consuming, so we do not include error bars in our main paper, following common practice in the community [1][2]. As requested, for more rigorous evaluation, we conducted multiple independent runs for the Qwen2.5-1.5B-Instruct and Qwen2.5-7B-Instruct models and report the results in Tables 4 and 5. The results demonstrate the stability and robustness of CPPO.
>
> Table 4: Comparison between GRPO and CPPO on the GSM8K test subset. We independently train Qwen2.5-1.5B-Instruct on the GSM8K training subset three times to calculate the mean and standard deviation, and the number of retained completions after pruning is $k = \left\lfloor G \times (1 - P) \right\rfloor$.
>
> |Method|Group Size ($G$)|Pruning Rate ($P$)|$k$|Accuracy (\%)|Training Time (s)|Accelerate Ratio|
> |-|-|-|-|-|-|-|
> |Qwen2.5-1.5B-Instruct|-|-|-|55.72|-|-|
> |GRPO|16|0.00\%|16|77.38 $\pm$ 0.28|23500.33 $\pm$ 130.49|1.00|
> |CPPO|16|50.00\%|8|78.15 $\pm$ 0.37|12862.33 $\pm$ 78.68|1.83 $\pm$ 0.01|
> |CPPO|16|75.00\%|4|78.76 $\pm$ 0.25|7436.00 $\pm$ 232.98|3.16 $\pm$ 0.08|
> |CPPO|16|87.50\%|2|80.01 $\pm$ 0.38|4516.33 $\pm$ 237.46|5.22 $\pm$ 0.31|
> |CPPO|16|93.75\%|1|78.99 $\pm$ 1.01|2946.00 $\pm$ 94.44|7.98 $\pm$ 0.23|
>
> Table 5: Comparison of GRPO and CPPO on the MATH test subset. We independently train Qwen2.5-7B-Instruct on the MATH training dataset three times to calculate the mean and standard deviation, and the number of retained completions after pruning is denoted by $k = \left\lfloor G \times (1 - P) \right\rfloor$.
>
> |Method|Group Size ($G$)|Pruning Rate ($P$)|$k$|Accuracy (\%)|Training Time (s)|Accelerate Ratio|
> |-|-|-|-|-|-|-|
> |Qwen2.5-7B-Instruct|-|-|-|55.20|-|-|
> |GRPO|16|0.00\%|16|75.26 $\pm$ 0.09|33795.00 $\pm$ 80.18|1.00|
> |CPPO|16|50.00\%|8|76.01 $\pm$ 1.03|20129.00 $\pm$ 298.95|1.68 $\pm$ 0.02|
> |CPPO|16|75.00\%|4|76.55 $\pm$ 0.83|13067.00 $\pm$ 81.26|2.59 $\pm$ 0.02|
> |CPPO|16|87.50\%|2|75.95 $\pm$ 0.55|9722.00 $\pm$ 78.87|3.48 $\pm$ 0.03|
> |CPPO|16|93.75\%|1|74.65 $\pm$ 1.31|7608.00 $\pm$ 542.36|4.46 $\pm$ 0.29|
>
> [1] Shao, Zhihong, et al. "Deepseekmath: Pushing the limits of mathematical reasoning in open language models." arXiv preprint arXiv:2402.03300 (2024).
> [2] Guo, Daya, et al. "Deepseek-r1: Incentivizing reasoning capability in llms via reinforcement learning." arXiv preprint arXiv:2501.12948 (2025).
>
> We hope that our responses can well address your concerns. If you have any further questions, please feel free to contact us. Thank you again for spending time reviewing our work.

---

### Official Review · Reviewer_o2Dn · 2025-07-04

**Clarity:** 3
**Significance:** 2
**Originality:** 2
**Rating:** 4
**Confidence:** 3

**Summary:**

This paper introduces Completion Pruning Policy Optimization (CPPO), a method to accelerate the training of reasoning models that use Group Relative Policy Optimization (GRPO). The key insight is that not all completions contribute equally to policy training—their contribution depends on their relative advantage. CPPO prunes completions with low absolute advantages before forward computation, thus reducing computational overhead. The authors also introduce a dynamic completion allocation strategy to maximize GPU utilization. Experiments on GSM8K and MATH datasets using Qwen2.5 models show speedups of up to 8.32x on GSM8K and 3.51x on MATH while maintaining or improving accuracy.

**Questions:**

I think this paper proposed a good intuitive solution, but the limited evaluation scope might not justify the benefit that this algorithm can bring. If the authors can demonstrate the robustness of this algorithm in a large variety of settings, it would help establish the value of this technique. For example:

- can you include comparison with other completion selection strategies?
- how does the method perform with larger models beyond 7B parameters?
- how to set hyperparameters for different settings? If different tasks want to adopt this technique, how do they choose $k$, $b$ and the pruning rate?

**Ethical Concerns:**

["NO or VERY MINOR ethics concerns only"]

**Final Justification:**

I appreciate the authors' answer to my questions and effort to conduct additional experiments to demonstrate the robustness of the approach for larger model and additional benchmark datasets. The results comparing and combining DAPO and CPPO is good to include in the main paper to provide additional insights for readers to understand the difference in these acceleration algorithms. Overall, I think the proposed method has its practical value and does bring additional benefits in training speed while maintaining comparable accuracy.

**Limitations:**

See questions

**Quality:**

3

**Strengths And Weaknesses:**

Strengths

The paper addressed a computational limitation in GRPO training, which needs to process multiple completions per question and scales training time multiplicatively. The problem formulation is clear and well-motivated. The proposed CPPO algorithm is intuitively motivated by decomposing the policy objective derivative and observing that advantage can be computed prior-forward, and this already existing information can be used to perform pruning. Paper also introduced a dynamic completion allocation strategy to maximize GPU utilization for practical mutli-GPU setting.

Two models are finetuneing with CPPO and evaluation on two datasets AMC and AIME shows good accuracy improvements over GRPO with low quality completions being removed during the training process. The analysis section provided some intuitive explanations why the proposed algorithm is working well.

Weakness

The core contribution of this paper is adopting a heuristic for selecting a subset of data points during training process. While some experiments show that this is effective, the insight that high-advantage completions contribute more to training is quite intuitive, and pruning low-advantage completions is a natural response to this observation. I feel it lacks some fundamental algorithmic innovation.

The evaluation section can use more experiment settings, such as testing CPPO's advantage with larger models (currently only <=7B parameters) as smaller models in general do not suffer as much in training inefficiency. There are many other math reasoning benchmark datasets, this paper only evaluated on two. There also do no seem to be comparisons made with other acceleration methods for RL training.

---

> ### Author Rebuttal · Authors · 2025-07-29
>
> Thank you for your valuable feedback, including remarks such as "The problem formulation is clear and well-motivated" and "a good intuitive solution." We give our point-by-point responses to your comments below:
>
> **Q1: Algorithmic innovation.**
>
> **A1:** With all due respect, we believe our work presents significant algorithmic innovation. GRPO is a widely used and influential reinforcement learning algorithm, but its substantial training overhead limits efficiency and scalability. Improving training efficiency is an important and practical problem. To our knowledge, this is the first study to accelerate GRPO by pruning completions, supported by rigorous theoretical analysis. In Appendix B, we provide detailed derivations, and in Section 3.2, we analyze the gradient of the GRPO policy objective to clarify the role of each term. Our analysis demonstrates that **not all completions contribute equally to the policy gradient**, and the contribution of each completion depends on its absolute advantage value. Based on this insight, we propose CPPO and dynamic completion allocation to speed up GRPO training by a considerable margin. All the theoretical analyses, insights, and methods are introduced for the first time in this paper. The novelty and theoretical soundness of our work are also recognized by reviewers ZKXt and kqfe.
>
> **Q2: More benchmark datasets.**
>
> **A2:** We evaluate models trained with GRPO and CPPO on the MATH training dataset and test them on several out-of-distribution benchmarks, including GPQA:Diamond, Olympiad_bench:zh, Agieval:gaokao-mathqa, MMLU:college_mathematics, and MathQA. As shown in Table 1, CPPO achieves comparable average performance to GRPO across these datasets, demonstrating that CPPO can accelerate GRPO training without compromising the model’s generalization ability. While we agree that evaluating our method on more benchmark datasets would be beneficial, limited academic computational resources restrict us to experiments on representative and high-quality datasets such as GSM8K, MATH, AMC, and AIME to validate the effectiveness of our method. We will continue to evaluate our method on additional benchmarks and tasks in future work.
>
> Table 1: Comparison between GRPO and CPPO on additional benchmark datasets. We train Qwen2.5-7B-Instruct on the MATH training dataset and evaluate the model on various out-of-distribution benchmarks using lighteval. The number of retained completions after pruning is given by $k = \left\lfloor G \times (1 - P) \right\rfloor$.
>
> |Method|Group Size ($G$)|Pruning Rate ($P$)|$k$|GPQA:Diamond (\%)|Olympiad_bench:zh (\%)|Agieval:gaokao-mathqa (\%)|MMLU:college_mathematics (\%)|MathQA (\%)|Average (\%)|
> |-|-|-|-|-|-|-|-|-|-|
> |Qwen2.5-7B-Instruct|-|-|-|33.46 $\pm$ 2.46|38.23 $\pm$ 1.38|45.58 $\pm$ 2.66|26.00 $\pm$ 4.41|33.47 $\pm$ 0.86|35.35|
> |GRPO|16|0.00%|16|33.71 $\pm$ 2.55|40.97 $\pm$ 1.40|49.00 $\pm$ 2.67|26.00 $\pm$ 4.41|34.30 $\pm$ 0.87|36.80|
> |CPPO|16|50.00%|8|34.60 $\pm$ 2.70|42.74 $\pm$ 1.41|47.59 $\pm$ 2.67|27.00 $\pm$ 4.46|34.71 $\pm$ 0.87|37.33|
>
> **Q3: Comparison with other acceleration methods for RL training.**
>
> **A3:** Our work aims to accelerate the GRPO algorithm, which is widely used and influential in reinforcement learning. Due to significant differences between GRPO and other reinforcement learning algorithms, such as PPO, it is not appropriate to compare our method with acceleration techniques designed for other algorithms. To our knowledge, this is the first work to accelerate GRPO by pruning completions. Therefore, we only compare our method with CPPO variants in Table 3 of the main paper. Recently, a concurrent work[1] has also sought to accelerate GRPO, but it focus on different aspects and is orthogonal to our work. As shown in Table 2, CPPO can be combined with it to further improve training time and accuracy. We will include these results in the camera-ready version.
>
> Table 2: Comparison between different reinforcement learning algorithms on the GSM8K test subset. We train Qwen2.5-1.5B-Instruct on the GSM8K training subset, and the number of retained completions after pruning is $k = \left\lfloor G \times (1 - P) \right\rfloor$. All experiments are conducted on the verl framework, which supports various RL algorithms.
>
> |Method|Group Size ($G$)|Pruning Rate ($P$)|$k$|Accuracy (\%)|Training Time (s)|Accelerate Ratio|
> |-|-|-|-|-|-|-|
> |Qwen2.5-1.5B-Instruct|-|-|-|55.19|-|-|
> |GRPO|16|0.00%|16|78.01|7741|1.00|
> |CPPO|16|50.00%|8|78.92|5192|1.49|
> |DAPO|16|0.00%|16|78.01|3800|1.00|
> |DAPO+CPPO|16|50.00%|8|78.01|2134|1.77|
>
> [1] Yu, Qiying, et al. "Dapo: An open-source llm reinforcement learning system at scale." arXiv preprint arXiv:2503.14476 (2025).
>
> **Q4: Comparisons with other completion selection strategies.**
>
> **A4:** As shown in Table 3 of the main paper, we compare our CPPO with other completion selection strategies. "Random" refers to randomly pruning completions. "Largest"/"Smallest" prune completions with the highest/lowest absolute advantages, respectively. "Largest*"/"Smallest*" prune completions with the highest/lowest advantages, which is equivalent to pruning completions with the highest/lowest rewards, since higher rewards correspond to higher advantages according to Eq.(3) of the main paper. The results in Table 3 of the main paper demonstrate that our CPPO, which prunes completions with the smallest absolute advantages, outperforms all other completion selection strategies.
>
> **Q5: Experimental results on larger models beyond 7B.**
>
> **A5:** As requested, we conduct experiments on larger models, specifically Qwen2.5-14B-Instruct, and present the results in Table 3. The results show that our CPPO can also accelerate the training of larger models by up to 2.83x without compromising accuracy, demonstrating the scalability and generalizability of our method. We would like to evaluate our method on even larger models, but due to limited academic computational resources, we can only conduct experiments on Qwen2.5-14B-Instruct at this time. We will include these results in the main paper.
>
> Table 3: Comparison of GRPO and CPPO on the MATH test subset. We train Qwen2.5-14B-Instruct on the MATH training dataset, and the number of retained completions after pruning is denoted by $k = \left\lfloor G \times (1 - P) \right\rfloor$.
>
> |Method|Group Size ($G$)|Pruning Rate ($P$)|$k$|Accuracy (\%)|Training Time (s)|Accelerate Ratio|
> |-|-|-|-|-|-|-|
> |Qwen2.5-14B-Instruct|-|-|-|67.80|-|-|
> |GRPO|16|0.00%|16|77.60|33942|1.00|
> |CPPO|16|50.00%|8|79.40|21375|1.59|
> |CPPO|16|75.00%|4|79.40|15385|2.21|
> |CPPO|16|87.50%|2|78.00|11997|2.83|
> |CPPO|16|93.75%|1|76.00|11247|3.02|
>
> **Q6: How to set hyperparameters for different settings? If different tasks want to adopt this technique, how do they choose $k$, $b$, and the pruning rate $P$?**
>
> **A6:** Assuming the group size is $G$ and the pruning rate is $P$, the hyperparameters are set as $k = \left\lfloor G \times (1 - P) \right\rfloor$ and $b = G / k$. Therefore, it is only necessary to choose appropriate values for $G$ and $P$ for different tasks. A larger group size $G$ generally improves performance but increases training cost, while a higher pruning rate $P$ provides greater speedup but may degrade performance if set too high. We recommend selecting $G$ and $P$ based on available computational resources and task performance requirements. In most cases, setting $G$ to 16 and $P$ to 50% achieves a good balance between performance and speedup.
>
> We hope our responses address your concerns. If you have any further questions, please feel free to contact us. Thank you again for spending time reviewing our work and providing valuable feedback.

---

> > ### Comment · Reviewer_o2Dn · 2025-08-08
> > **response to author rebuttal**
> >
> > I appreciate the authors' answer to my questions and effort to conduct additional experiments to demonstrate the robustness of the approach for larger model and additional benchmark datasets. The results comparing and combining DAPO and CPPO is good to include in the main paper to provide additional insights for readers to understand the difference in these acceleration algorithms. Overall, I think the proposed method has its practical value and does bring additional benefits in training speed while maintaining comparable accuracy. I have raised my score.

---

### Note · Authors · 2025-08-13

Dear Program Chairs, Area Chairs, and Reviewers,

We sincerely thank you for the time and effort you have dedicated to reviewing our paper.

In this paper, we introduce CPPO, a method that accelerates the training of reasoning models based on GRPO by pruning completions with low absolute advantages and using a dynamic completion allocation strategy to optimize GPU usage. Our approach is highly appraised by reviewers: Reviewer o2Dn noted that "the proposed method has its practical value"; Reviewer kqfe described it as "genuinely novel" and "well-motivated by the analysis of completion contributions"; and Reviewer ZKXt stated that "CPPO is a novel method that tackles a significant problem of training efficiency of GRPO."

During the rebuttal and discussion period, we have focused on addressing the reviewers' concerns and providing clarifications to further strengthen our paper. Specifically, in response to the major concern regarding the generalization of CPPO to larger models (o2Dn and zbVH), the need for more benchmarks (o2Dn and zbVH), additional reinforcement learning variants (kqfe), and different architectures (kqfe), we have provided: (1) results on Qwen2.5-14B-Instruct; (2) results on six additional benchmarks; (3) results combining CPPO with DAPO and Dr.GRPO; and (4) results on Llama-3.2-1B-Instruct. Crucially, all these new experiments consistently demonstrated that CPPO maintains its significant training speedup while achieving comparable or even better performance, confirming its effectiveness and generalizability.

To address concerns about the robustness and authenticity of our results (zbVH and ZKXt), we have included the mean and standard deviation for our main results. In response to questions about comparisons with other reinforcement learning algorithms (o2Dn and ZKXt), we have provided results comparing CPPO with REINFORCE++, PPO w/o KL, and DAPO. Finally, we have also clarified the algorithmic innovations, completion selection strategies, experimental settings, code release plans, stopping criteria, and addressed any potentially inappropriate statements. All these experiments, revisions, and analyses will be included in our final paper.

We believe these additional experiments and clarifications have sufficiently addressed the reviewers' concerns. We hope this summary assists the AC in the final decision. Thank you once again for your valuable time and consideration.

Sincerely,
Authors

---

### Decision · Program_Chairs · 2025-09-17

**Decision:**

Accept (poster)

**Comment:**

This paper presents an approach to reduce training time for RL runs with LLMs. This is done by dynamically allocating compute to sampling and training. The idea is to remove some completions that were not providing high magnitude advantages totally from training. The experiments shown on simplistic benchmarks or only on some models show gains in reducing training time.

The reviewers were generally positive about the paper, and liked it after the rebuttal. However, I think that a limitation of this paper is in not having a clear way to show benefits on multiple base models, for instance would the strategy help everywhere?